# Functional Association between Storage Protein Mobilization and Redox Signaling in Narrow-Leafed Lupin (*Lupinus angustifolius* L.) Seed Germination and Seedling Development

**DOI:** 10.3390/genes14101889

**Published:** 2023-09-28

**Authors:** Julia Escudero-Feliu, Elena Lima-Cabello, Esther Rodríguez de Haro, Sonia Morales-Santana, Jose C. Jimenez-Lopez

**Affiliations:** 1Department of Stress, Development and Signaling in Plants, Estacion Experimental del Zaidin, Spanish National Research Council (CSIC), 18008 Granada, Spain; julia.escudero@eez.csic.es (J.E.-F.); elena.lima@eez.csic.es (E.L.-C.); esther.rodriguez@eez.csic.es (E.R.d.H.); 2Proteomic Research Unit, Biosanitary Research Institute of Granada (ibs.Granada), 18012 Granada, Spain; soniamoralessantana@hotmail.com; 3The UWA Institute of Agriculture, The University of Western Australia, Crawley, Perth 6009, Australia

**Keywords:** oxidative stress, seed storage proteins, seed germination, conglutins, lupin

## Abstract

(1) Background: Seed storage mobilization, together with oxidative metabolism, with the ascorbate–glutathione (AsA-GSH) cycle as a crucial signaling and metabolic functional crossroad, is one of the main regulators of the control of cell morphogenesis and division, a fundamental physiological process driving seed germination and seedling growth. This study aims to characterize the cellular changes, composition, and patterns of the protein mobilization and ROS-dependent gene expression of redox metabolism in *Lupinus angustifolius* L. (narrow-leafed lupin, NLL) cotyledons during seed germination. (2) Methods: We performed gene expression analyses via RT-qPCR for conglutins α (1, 2, and 3), β (1, 2, and 5), γ (1, 2), and δ (2 and 4), including a ubiquitin gene as a control, and for redox metabolism-related genes; GADPH was used as a control gene. A microscopic study was developed on cotyledon samples from different germination stages, including as IMB (imbibition), and 2–5, 7, 9, and 11 DAI (days after imbibition), which were processed for light microscopy. SDS-PAGE and immunocytochemistry assays were performed using an anti-β-conglutin antibody (Agrisera), and an anti-rabbit IgG Daylight 488-conjugated secondary antibody. The controls were made while omitting primary Ab. (3) Results and Discussion: Our results showed that a large amount of seed storage protein (SSP) accumulates in protein bodies (PBs) and mobilizes during germination. Families of conglutins (β and γ) may play important roles as functional and signaling molecules, beyond the storage function, at intermediate steps of the seed germination process. In this regard, metabolic activities are closely associated with the regulation of oxidative homeostasis through AsA-GSH activities (γ-L-Glutamyl-L-cysteine synthetase, NOS, Catalase, Cu/Zn-SOD, GPx, GR, GS, GsT) after the imbibition of NLL mature seeds, metabolism activation, and dormancy breakage, which are key molecular and regulatory signaling pathways with particular importance in morphogenesis and developmental processes. (4) Conclusions: The knowledge generated in this study provides evidence for the functional changes and cellular tightly regulated events occurring in the NLL seed cotyledon, orchestrated by the oxidative-related metabolic machinery involved in seed germination advancement.

## 1. Introduction

Oxidative metabolism serves as a pivotal mechanism in the process of seed germination and subsequent plant development. Its activation serves a dual purpose: firstly, it facilitates the transition of a seed from its dormant state to an active one, and secondly, it regulates the progression through various stages of germination. In fact, the initiation of seed germination hinges on the presence of a specific amount of reactive oxygen species (ROS), which enables oxidative signaling without causing detrimental damage. This optimal range is referred to as the “oxidative window for germination” [1,2]. The accumulation of ROS during seed germination can occur through either non-enzymatic or enzymatic pathways. This underscores the intricate role of oxidative metabolism in orchestrating the complex process of seed germination [1,2].

During the terminal phase of seed maturation, when seeds are in a dry or desiccated state, which is a prevalent phenomenon in orthodox seeds, the generation of ROS primarily arises from non-enzymatic reactions. These reactions are an inevitable outcome of aerobic metabolism, including processes like lipid peroxidation [3,4,5,6]. In a desiccated state, the presence of ROS is a spontaneous result of encountering severe drought stress, a necessary factor for breaking dormancy. As a result, when the seeds absorb water and undergo the transition to a hydrated state during imbibition, their metabolic activities resume. This resurgence triggers a balance in ROS levels, a phenomenon known as ROS homeostasis. This equilibrium in ROS concentrations can potentially result from various cellular compartments, including the catabolic breakdown of lipids occurring within glyoxysomes [4,7,8,9,10]. Among ROS, the most prevalent compounds include hydrogen peroxide (H_2_O_2_), superoxide (O_2_^−^), and hydroxyl radical (HO·). These compounds have been observed to be generated during the germination of seeds across various species [11,12]. The diversity in the sources of ROS production and their ability to move is interconnected with the viscosity of the cytoplasm [13]. In their desiccated state, ROS primarily impact nearby targets due to restricted movement, whereas in their hydrated state, the presence of free water facilitates the extensive mobility of these oxidative species. This heightened mobility enables them to serve as potent signaling molecules or carriers of environmental cues throughout the process of seed germination [1,2]. Indeed, ROS could function as pivotal signals within hormonal networks that steer dormant seeds towards a state of dormancy release (as seen in after-ripened seeds), particularly when the seeds are in a desiccated state, indicative of dormancy alleviation.

The process of oxidative seed germination, particularly concerning seed desiccation and aging, is intricately associated with the detrimental effects of reactive oxygen species (ROS). Seed desiccation tolerance plays a crucial role in seed maturation as an adaptive mechanism. It enables seeds to survive prolonged storage, withstand adverse environmental conditions, and ultimately produce seeds with a robust germination capacity [11,14]. In orthodox seeds, desiccation tolerance to water absence is associated with an accumulation of free radicals, the synthesis of specific proteins, and the activation of antioxidant defense systems [15,16,17]. To regulate this homeostasis in the desiccation state, plants undergo adjustments in the functioning of antioxidant enzymes that scavenge ROS compounds. These enzymes encompass superoxide dismutase (SOD), catalase (CAT), and the enzymes of the ascorbate–glutathione (AsA-GSH) cycle. Activation of these enzymes occurs within the defined oxidative window of germination [18]. For example, some studies have demonstrated that enhancing drying tolerance leads to an upsurge in H_2_O_2_ production, concurrently modulating the expression of the *CAT* gene in sunflowers. In contrast, the bean plant (*Phaseolus vulgaris*) showcases an augmentation in CAT activity and a reduction in SOD and ascorbate peroxidase (APX) activities [19,20].

To overcome the stress, seed antioxidant activities have been demonstrated to prevent excessive ROS accumulation and oxidative damage [13,21]. However, these functions are diminished or even compromised within dormant seeds, contributing to the generation of ROS, which play a role in the release of seed dormancy [12,22,23]. For that reason, ROS are also fine-tuning signals in the germination process in the interaction with hormone signaling pathways. In addition, ROS are also exogenous stimulants for germination, as several studies have reported that exogenous H_2_O_2_ promotes seed germination in many plants [24,25]. Several investigations have indicated that the imbibition of pea seeds with H_2_O_2_ expedites germination and fosters the initial development of seedlings. In essence, this oxidative stress triggers a boost in antioxidative defenses, including the activation of ascorbate-oxidizing enzyme activity. This enzyme activity is of significant consequence in the context of scavenging ROS to eliminate injurious oxygen species that emerge as metabolic byproducts within cells or as outcomes of environmental pressures [26,27].

Modifications in redox components in oxidative stress can also alter proteins and nucleic acids during germination. In terms of proteins, the amino acids Cys (cysteine) and Met (methionine) exhibit pronounced susceptibility to oxidation. For instance, the transformation of thiol groups into disulfide bonds is a recognized redox regulatory process. Additionally, carbonylation can lead to functional impairment and, in some cases, the degradation of proteins [28,29]. Indeed, protein carbonylation is a modification that has been evidenced to be important for the mobilization of seed storage proteins (SSP) as it increases protein susceptibility towards proteolytic cleavage by the 20S proteasome, as is shown in *Arabidopsis thaliana*, rice, and pea seeds [30,31,32,33]. In rice, the carbonylation of cupin family of proteins in 0 h embryos promoted early germination, facilitating their mobilization, activating pentose phosphate, and blocking the glycolysis biochemical pathway [32]. SSP mobilization, and therefore carbonylation, is a crucial physiological process in a plant’s life cycle in regulating the breaking of seed dormancy [34]. SSPs in dicotyledonous plants accumulate in specialized storage cells in embryonic tissues (cotyledon and endosperm), being protected from cleavage degradation by internal proteases. In the same way, mature SSPs are deposited into two specialized compartments protected from cytoplasmic proteases: (i) protein bodies (PB) and (ii) protein storage vacuoles (PSVs), respectively. Protein bodies (PBs) are specialized plant cell organelles primarily dedicated to the storage of seed storage proteins [35]. They originate during seed maturation through the division of vacuoles and serve as vital protein storage units in the seed development process. In contrast, protein storage vacuoles (PSVs) are larger organelles found in various plant cell types. They also store proteins, albeit not exclusively in seeds. PSVs have a broader cellular role, encompassing the storage of enzymes and other cellular components. Unlike PBs, PSVs are not exclusive to seed maturation [35,36]. SSP turnover is a key event in nourishing the seedlings. Overall, storage proteins are synthesized in the rough endoplasmic reticulum (ER) and are co-translationally targeted to the ER lumen where they undergo several modifications and storage in PBs. Eventually, these PBs result from the division of the vacuoles during seed development and the maturation process [35]. In olive (*Olea europaea* L.), seed storage legumin-like proteins mobilization has been enhanced by different degradation patterns, leading to amino acid accumulation for seed germination [36]. Moreover, this mobilization was shown to have different and specific spatial and temporal patterns in cells and tissues and vary depending on the seed species.

Two major fractions of SSPs constituted mainly 11S globulins (legumin-like protein) and 7S-globulins (vicilin proteins) and can vary in an intra- and interspecific manner [37]. In NLL plants, lupin seed globulins comprise four families: α-conglutin (11S) and β-conglutin (7S) as major components, and γ- conglutin (7S) and δ-conglutin (2S) as a minor components [38]. NLL β-conglutins are characterized by multiple agricultural benefits such as the protective function of seed β-conglutin proteins against necrotrophic pathogens attack [39]. They are also characterized by their adaptation to drought stress and seed desiccation tolerance. In consequence, numerous events must occur at the cellular and biochemical levels, such as the synthesis and accumulation of SSPs and their appropriate mobilization [38]. Despite the importance of these physiological cues for breaking dormancy, leading the seed to germinate, little information is known about the regulatory features underlying the biochemical, molecular, and signaling processes taking place during NLL seed germination [40]. Indeed, many questions must be resolved regarding the cellular signaling mechanisms and ROS transcriptome signature specific for potential modulation of seed development [41]. This study aims to characterize (1) the cellular changes that occur during and after NLL germination; (2) the pattern of protein mobilization; and (3) the extent of redox metabolism changes, as a regulator of ROS-dependent signaling driving NLL seed germination and seedling growth.

In recent years, studies have shed light on the intricate processes underlying seed germination and seedling development in plants, with a particular focus on NLL. Notably, research efforts have elucidated the pivotal role of storage proteins, such as conglutins, in these developmental stages. Czubinski and Feder emphasized the significance of conglutins in providing a crucial nitrogen source during seedling growth, underscoring their contribution to the plant’s nutrient reservoir, in a recent paper [42]. Concurrently, Cabello-Hurtado et al. conducted a comprehensive proteomic analysis, revealing dynamic changes in protein mobilization during lupin seed development. Their findings accentuated the pivotal role of conglutins in provisioning the energy and nutrients essential for seedling vigor [43]. Additionally, Gulisano et al. also explored the intricate redox regulation that occurs during lupin seed germination, implicating oxidative stress and antioxidant defense mechanisms [44]. These studies collectively suggest that conglutins and other storage proteins may interface with redox signaling pathways, contributing to the orchestration of seed germination and early seedling development [42,43,44]. Furthermore, Tahmasian et al. illuminated the nutritional aspects of lupin seeds in their recent investigation from 2022, highlighting NLL conglutins as a source of essential amino acids and bioactive compounds [45]. Together, this body of literature underscores the multifaceted role of conglutins in lupin seed biology and provides a foundation for our investigation into the functional association between storage protein mobilization and redox signaling in *Lupinus angustifolius* L. seed germination and seedling development [42,43,44,45].

## 2. Materials and Methods

### 2.1. Microscopy Study of NLL Cotyledon during Seed Germination

#### 2.1.1. Imbibition and Germination of NLL Seeds

The specie used for the conducted studies was *L. angustifolius*, and the seeds were provided by The University of Western Australia. Seed imbibition was achieved by hydrating the seeds for 24 h on sufficiently moist filter paper. After this, the seeds were allowed to germinate in MS medium (Murashige and Skoog Medium, specifically used for the germination of NLL to compare germination with and without this medium) or in H_2_O under sterile conditions. MS medium contains the following mineral salts: NH_4_NO_3_ 0.47 g/L, KNO_3_ 1.31 g/L, MgSO_4_·7H_2_O 0.24 g/L, KH_2_PO_4_ 0.13 g/L, KCl 0.074 g/L, Ca(NO_3_)_2_·5H_2_O 0.59 g/L, NaFe-EDTA 10 mL/L, Micronutrients: MnSO_4_·H_2_O 1.690 g/L, ZnSO_4_·7H_2_O 0.860 g/L, CuSO_4_·5H_2_O 0.0025 g/L, KI 0.0830 g/L, CoCl_2_·7H_2_O 0.0025 g/L, H_3_BO_3_ 0.620 g/L, Na_2_MoO_4_·2H_2_O 0.025 g/L, Sucrose 2 g/L, and myo-inositol 0.1 g/L.

Photographs of the seeds were taken every 24 h, and a portion of this material was stored at −80 °C for subsequent analysis. The final stage studied was 240 h after imbibition. The stages studied during germination included 24 h of imbibition (stage 1, 1 DAI), 24 h of germination (stage 2, 2 DAI), 48 h (stage 3, 3 DAI), 72 h (stage 4, 4 DAI), 96 h (stage 5, 5 DAI), 144 h (stage 6, 7 DAI), 192 h (stage 7, 9 DAI), and 240 h (stage 8, 11 DAI).

#### 2.1.2. Preparation of NLL Seed Samples for Light Microscopy

Following the germination of blue lupin seeds, small cotyledon fragments were dissected and fixed for 24 h at 4 °C using a fixation solution. Subsequently, the samples underwent a dehydration process using alcohol solutions with progressively increasing concentrations of 50%, 70%, 96%, and 100%, and finally, xylene. After dehydration, the samples were embedded in Unicryl plastic resin (BBInternational, Montebelluna, Italy) at increasing concentrations of Unicryl/ethanol, and then, subjected to polymerization using exposure to ultraviolet light at −20 °C for 72 h. Semithin sections (≈1 μm) of these samples were obtained using a Reichert-Jung Ultracut E ultramicrotome (Leica Microsystems, Wetzlar, Germany). The semithin sections were placed on glass slides coated with BioBondTM adhesive film (BBInternational, Montebelluna, Italy).

#### 2.1.3. Observation of NLL Seed Samples via Light Microscopy

For general sample observations, the sections were stained with a mixture of 0.05% (*w*/*v*) methylene blue and 0.05% (*w*/*v*) toluidine blue. In both cases, the sections were stained for 5 min at room temperature and washed with running water for 2 min. The samples were observed using a Nikon C1 confocal laser scanning microscope (Nikon, Konan, Minato-ku, Tokyo, Japan) with an argon laser (488 nm) in bright-field mode. The images were captured and processed using EZ-C1 Gold software version 2.12, build 240 (Nikon, Konan, Minato-ku, Tokyo, Japan).

In order to discern the distribution of lipids within the cotyledon samples, Nile Red staining was employed. This lipid-specific staining allowed for the visualization of lipid-rich regions, shedding light on potential lipid accumulation and mobilization during different germination stages.

#### 2.1.4. Immunocytochemical Analysis

This experiment had the aim of investigating the morphological and also immunocytochemical changes occurring in the cotyledons of NLL seeds during the germination process, for which a microscopy study was performed. Immunocytochemical investigations were conducted to determine the presence and distribution of β-conglutin proteins within the cotyledon tissues. To accomplish this, an anti-β-conglutin antibody developed in-house (custom-made, Agrisera, Vännäs, Sweden) following Jimenez-Lopez et al. [39], which is a rabbit polyclonal antiserum, was used as the primary antibody. Subsequently, an anti-rabbit IgG Daylight 488-conjugated secondary antibody was employed to enable the visualization of the primary antibody complexes. To ensure the specificity and reliability of the immunocytochemistry assay, control experiments were executed. These control assays involved the omission of primary antibodies, thereby serving as a reference to validate the binding specificity of the primary antibodies to their target antigens. For fluorescence microscopy, the samples were visualized using a confocal laser scanning microscope (Nikon C1) equipped with an argon laser (488 nm) and bright-field capabilities. This allowed for precise localization and visualization of the β-conglutin proteins.

### 2.2. Gene Expression Analyses during the NLL Seed Germination Process

#### 2.2.1. Total RNA Isolation and cDNA Amplification

Total RNA was extracted from germinating NLL seeds using the RNeasy Plant Total RNA kit (Qiagen, Hilden, Germany), following the manufacturer’s recommended protocols. Subsequently, first-strand cDNA synthesis was performed using Superscript II reverse transcriptase (Invitrogen, Waltham, MA, USA), employing a poly-dT adaptor as a primer. This cDNA amplification process was carried out in accordance with the manufacturer’s instructions.

#### 2.2.2. Real-Time PCR Analysis

For the assessment of gene expression dynamics, real-time PCR was conducted using the QuantiTec SYBR Green PCR master mix (Applied Biosystems, Waltham, MA, USA). Specific forward and reverse primers were designed for the targeted conglutins [α (1, 2, and 3), β (1, 2, 5), γ1, δ (2, and 4)], as well as for the ubiquitin reference gene. Furthermore, a comprehensive analysis of genes associated with reactive oxygen species (ROS) metabolism was undertaken, with an additional control group (designated as ‘G’). The primer sequences used for RT-qPCR are shown in Table 1 and Table 2. Additionally, in our experimental design, we ensured sample distribution by triplicating the reactions for the genes of interest on each plate and by repeating the experiments at least two or three times in order to ensure a robust dataset for the performance of statistical analysis.

#### 2.2.3. Data Collection and Controls

Each real-time PCR assay was thoughtfully assembled to include essential controls, comprising a template-omitted control and a reverse transcription (RT) negative control. These controls served to validate the accuracy and reliability of the subsequent analyses. By meticulously executing these methodological steps, we aimed to unravel the intricate interplay of oxidative gene expressions throughout the dynamic process of narrow-leaved lupin seed germination. The integration of total RNA isolation, cDNA synthesis, and precise real-time PCR analyses constituted a robust approach for comprehensively understanding the genetic responses associated with ROS metabolism and conglutin gene expression during NLL seed germination.

#### 2.2.4. Data Analysis

Changes in gene expression levels (shown in Table 1 and Table 2, respectively) were determined using the formula 2^−Δ(ΔCt)^ [46], where the cycle threshold (CT) at which transcripts were detected was normalized to the CT at which the constitutive gene *Glyceraldehyde-3-phosphate dehydrogenase* (*GAPDH*) or *ubiquitin* (*UBQ*), was detected, referred to as ΔCT. The PCR efficiency was determined through an analysis based on standard curves for the amplification of the selected genes and the endogenous control, which exhibited high reproducibility.

### 2.3. Biochemical Analysis of β-Conglutin Proteins during NLL Seed Germination Process

The protein profile of β-conglutin proteins during the seed germination process of NLL was investigated through a series of well-defined biochemical steps.

-SDS-PAGE separation: Initially, protein samples containing 10 µg of protein per sample were subjected to electrophoresis on 4–20% Mini-PROTEAN^®^ TGX™ Precast Gels (Bio-Rad, Hercules, CA, USA), utilizing the Mini-PROTEAN^®^ Tetra Cell apparatus (Bio-Rad). Following electrophoresis, the proteins were visualized using Coomassie Brilliant Blue staining following established protocols.-Transfer onto PVDF membrane: To facilitate further analysis, the proteins were subsequently transferred from the gel onto polyvinylidene fluoride (PVDF) membranes using the Mini-Trans-Blot Electrophoretic Transfer Cell system (Bio-Rad). This step allowed for the subsequent immunodetection of specific proteins.-Blocking and primary antibody incubation: The PVDF membranes were blocked for 2 h with a blocking solution containing 5% (*w*/*v*) non-fat dry milk in Tris-buffered saline (TBS) buffer at pH 7.4. The immunodetection of β-conglutin proteins was achieved by incubating the membranes with a rabbit polyclonal antiserum developed in-house. The antiserum was diluted to a ratio of 1:1000 in TBS buffer containing 5% (*w*/*v*) non-fat dry milk and 0.5% Tween-20.-Secondary antibody and signal detection: Following primary antibody incubation, a secondary antibody, horseradish peroxidase (HRP)-conjugated anti-rabbit IgG (Bio-Rad), was utilized at a dilution ratio of 1:3000 in TBS buffer with 0.5% Tween-20. The secondary antibody was incubated for 2 h, followed by three 15 min washing steps with TBS containing 0.5% Tween-20 to remove any unbound antibodies.-Chemiluminescence detection: The presence of β-conglutin proteins was detected through chemiluminescence. The membranes were exposed to the SuperSignal^®^ West Pico Chemiluminescent substrate (Thermo Scientific, Waltham, MA, USA), and the resulting chemiluminescent signal, indicative of the presence of β-conglutin proteins, was captured on X-ray films (Kodak, Rochester, NY, USA). This comprehensive procedure allowed for the detailed examination of β-conglutin proteins and their variations throughout the Narrow-Leafed Lupin seed germination process.

This biochemical analysis provided crucial insights into the dynamics of β-conglutin proteins during seed germination, further enhancing our understanding of this intricate biological process.

### 2.4. Statistical Analysis

All experiments were performed at least in duplicate or triplicate and the results were expressed as mean ± standard deviation unless otherwise indicated. Statistical analyses were performed using the Shapiro–Wilk test to analyze the normality of the data set and One- or Two-Way ANOVA analysis, with Dunnett or Tukey correction, depending on the number of groups and data of each experiment, using GraphPad Prism 9 software, version 9.3.0. Statistical differences between samples were considered significant when *p* values were *p* < 0.05 (*), *p* < 0.01 (**), or *p* < 0.001 (***).

### 2.5. Bioinformatic Analysis

#### 2.5.1. Genetic Resources

The expression analysis of genes was sourced from the recently sequenced transcriptome of *Lupinus angustifolius* L. These sequence data were design on the basis of data provided through a collaborative research effort in the narrow-leafed lupin genome project, https://lupinexpress.org/ (access on 26 September 2023). Identification of all the analyzed genes was accomplished using the BLAST tool, and we compared them with genes from the same families in model organisms such as *Arabidopsis* and *Medicago*.

#### 2.5.2. Primer Design

Primers for the qPCR assays were designed using the bioinformatic tool PrimerQuest. The design process took into consideration factors such as base pair length similarity, percentage of GC pairs, and hybridization temperature. These primers were subsequently synthesized by Invitrogen (Thermo Fisher, Waltham, Massachusetts, USA).

## 3. Results and Discussion

### 3.1. Microscopy Study of PBs during Lupin Seed Germination Reveals the Presence of β-Conglutins

In this first experiment, a microscopy study of PBs during lupin seed germination was performed following the method outlined in Section 2.1., in order to comprehensively investigate the morphological and immunocytochemical changes occurring in the cotyledons of NLL seeds during the germination process. Figure 1A–C illustrate discernible morphological variations among the cells, accompanied by distinct patterns in the distribution of storage components within the cotyledon tissue. Furthermore, the application of the immuno-histochemistry assay unveiled histochemical attributes in the cotyledon (Figure 1E). Notably, control images for all the examined samples exhibited negligible auto-fluorescence signals (Figure 1D). It is noteworthy that our investigation exclusively focused on the cotyledon, in contrast to analogous studies conducted on olive seeds [36,47]. This distinction arises from the absence of an endosperm in lupin seeds due to its degradation and subsequent absorption by the developing embryo.

Regarding Figure 1A,B, at the initial imbibition (IMB) stage, protein bodies (PBs) within the cells exhibit a substantial presence of storage proteins. Subsequently, these PBs undergo a process of content depletion, leading to the creation of voids known as “holes” (H). This phenomenon is a consequence of protein degradation, where the released proteins serve as vital nutrients for facilitating the germination and subsequent growth of the plant [48]. Our results showed that a large amount of storage protein is stored in the PB and mobilized during germination (Figure 1). At the IMB stage, protein accumulation is higher in PBs, and as imbibition progresses (2 to 11 DAI), the degradation of these proteins occurs. In this way, the proteins are subsequently consumed for seed germination, generating remnants known as “protein remains” (PR), which are positioned at the periphery of these PBs and, over time, transform into empty protein bodies (EPB), which then coalesce until they ultimately vanish.

Figure 1C showcases the utilization of the lipid-specific Nile Red staining technique. Notably, a lipid droplet is indicated by an arrowhead in this staining. The distribution of Lipid Bodies (LBs) within the cotyledon predominantly encompasses the periphery of protein bodies (PBs), as depicted in Figure 1C. This arrangement gives rise to distinct cytoplasmic spaces that are readily observable. Furthermore, as the imbibition progresses, there is a gradual disappearance of lipid droplets, a phenomenon congruent with the concurrent degradation of PBs during this process.

Figure 1E underscores the immunocytochemical localization of β-conglutins. Cellular remains (CR) are discernible, alongside the cell wall (CW), within the depicted context. The structural elements of protein bodies (PBs) are evident. In contrast, Figure 1D portrays the outcome of the control assay, highlighting the absence of significant autofluorescence signals across all examined samples.

As is demonstrated in mature olive seeds, SSP mobilization sites often coincide with embryonic axes where PBs are found in the cytoplasmic space. The mobilization of these storage compounds occurs diachronically in a specific spatial and temporal pattern, governing seed germination rates [36].

According to the investigations of Jimenez-Lopez et al., 2013 [36], olive cotyledon tissues showed an irregular distribution and a highly variable number of lightly dense PBs of 2–25 um of diameter. Differences were also observed in the immunochemistry assays, with a negative correlation between the PB area and the intensity of immunofluorescence. Following this research, the authors demonstrated an intense signal reflecting the beginning of the protein mobilization process in PBs at 15 h of imbibition, similarly to our results at the IMB stage. Further steps of germination revealed reorganization and PBs’ fusion, leading to a large vacuole occupying the whole cytoplasm at 4 DAI and three different patterns from 5 to 10 DAI. Therefore, PBs result in (i) holes, (ii) tree-like structures, and (iii) complete disintegration of the PBs, when analyzed the olive seed tissues. Not until the seeds reached 11 DAI did the cytoplasm of these cells display clusters of proteins of varying numbers, sizes, and shapes, regardless of the mobilization pattern [36]. This study, and another similar one by the same authors in olive cotyledon [47], described, for the first time, a comprehensive account of the three distinct pathways by which SSPs are mobilized within the PBs of cotyledon cells. These pathways of seed protein mobilization may concurrently take place within the same cell type, yet it remains plausible that distinct mobilization pathways operate in diverse cell types within the cotyledon. This prospect is consistent with the observation that the distinctive enzymatic composition of lytic vacuoles can be synthesized in a tissue-specific manner during development. This phenomenon might exert a pivotal influence in determining the specific route of SSPs mobilization [47,49].

Like our results, and as already described, the PBs in NLL seeds disappeared throughout the days of imbibition, as can be seen in the analysis of both the general staining and immunohistochemical localization of β-conglutins (Figure 1A, 1B and 1E). PB lupin storage proteins, according to our analysis, leave (i) holes, (ii) PR, and (iii) EPB that coalesced into a large vacuole until they disappeared. This partially matches the previous findings in olive cotyledon [36,47]. The intrinsic features of SSP mobilization and degradation suggest that proteolytic enzymes originate within protein masses and accumulate inactively along with SSPs awaiting further activation. Another example has been shown in soybean mobilization, where storage proteins such as glycinin and β-conglycinin proteolysis result in the appearance of intermediate protein products, which will eventually be degraded during seedling growth [50].

In order to analyze the time-course of SSP synthesis and compare it with the previous observations at different developmental stages of the NLL cotyledon during seed germination, we performed protein extractions at the different stages (IMB and DAIs) to detect NLL legumin proteins via immunoblot assays. The results are shown in Section 3.2**.**

### 3.2. Biochemical Analysis of β-Conglutins during NLL Seed Germination Process

As mentioned in *Materials and Methods*, the protein profile of β-conglutin proteins during the seed germination process was assessed following the protocol outlined in Section 2.3. The results are shown in Figure 2.

Overall, the protein content decreased as the days of imbibition passed (1: IMB, 2–8: subsequent days after imbibition (DAI) stages—2, 3, 4, 5, 7, 9, and 11, respectively, designated as numbers 2 to 8). Considering the amount of total seed protein and the β-conglutin immunoblotting, a tendency toward degradation was observed throughout the DAI. This analysis shows an accumulation between the values of 50 and 75 kDa corresponding to β-conglutin proteins molecular weights (MW), and even degradation in smaller ones with lower values from 50 kDa at 7 DAI and disappearance at 8 DAI (Figure 2, upper panel). The immunoblotting signal displayed is between 50 and 75 kDa, which is consistent with the signals obtained in the other assays, which were shown to be around 70 kDa (Figure 2, lower panel) [39].

These results are in concordance with the microscopic analysis of the seed (Figure 1), as from 8 DAI, the reserve protein content was diminished, generating large vacuoles that filled the whole cell cytoplasm. Likewise, the reserve protein content was low at 1 DAI as the reserve proteins had not developed sufficiently in the PBs. The most progressive and remarkable diminishment occurred from the 6th day of the germination process onward.

In essence, the coming together of findings from our protein analysis, microscopic observations, and the stages of seed protein development supports a clear pattern of how proteins degrade and transform during seed germination. This process is intricately linked with the overall process of seed growth.

### 3.3. Oxidative Metabolism Promotes the Mobilization of Reserve Proteins and Seed Germination

#### SSPs Are Degraded during Seed Maturation

Further analysis of SSP expression was performed throughout germination to see how the expression of each of the families of globulins, α-conglutin (or 11S legumin), β-conglutin (or 7S vicilin), γ-conglutin (or 7S-basic globulin), and δ-conglutin (or 2S albumins) varied. As for the other analyses, the largest conglutin family was β (56%), followed by α (24%), δ (15%), and γ (6%) [51]. The results of the gene expression analyses during the NLL seed germination process are shown in Figure 3.

In our analysis, the different isoforms of each conglutin were investigated. For the α-conglutin family, a general pattern is observed among the three isoforms during germination, as decreased expression was observed throughout the DAI. 1α-conglutin transcription showed higher expression throughout the imbibition steps compared to the α2 and α3 isoforms, whose expression values dropped drastically from 5 DAI. This may be due to the fact that isoforms α2 and α3 are phylogenetically closer to each other than α1-conglutin [51]. Furthermore, this transcription pattern was also observed for isoforms γ1 and γ2-conglutin and 1δ and 4δ-conglutin. However, their expression was higher during the early germination stages of γ2 conglutin, and this expression decreased by 6 DAI compared to that of the γ1-conglutin isoform. Similar behavior was reported for 4δ-conglutin, which dropped to 5 DAI compared to δ1-conglutin, which did so at 7 DAI, as observed in Figure 3.

For the β-conglutin protein family, the transcription pattern varied between isoforms, showing a tendency toward conglutin degradation throughout germination, except for β4 and β6. These two specific scenarios portray an alternate transcriptomic landscape characterized by protein transcription values that surge at later germination stages. For β4, this peak expression notably surpassed the 5-day mark after imbibition (DAI), while for β6, the peak occurred around 3 DAI. Conversely, in the case of the β7-conglutin isoform, a conspicuous absence of expression was observed throughout the imbibition stages in comparison to the control at 1 DAI. By delving into these intricate transcriptional profiles, it becomes evident that certain isoforms, specifically conglutins β4 and β6, deviate from the conventional trend exhibited by storage proteins. These patterns of expression hint at a functional role distinct from typical storage proteins, offering a glimpse into the multifaceted roles that conglutin proteins may undertake during the process of germination [38,39,40].

Concretely, it is very interesting to compare the obtained results of our comprehensive analysis of conglutin gene expression during NLL seed germination with the existing knowledge in this field. Foley et al. (2015) previously investigated conglutin gene expression, and by comparing their findings to our results, several intriguing distinctions emerge [38]. One prominent difference between both results is evident in the expression dynamics of α-conglutin isoforms. While Foley et al. reported a general decrease in α-conglutin expression during germination, our results indicated nuanced variations among α-conglutin isoforms. Specifically, isoform α1-conglutin exhibited higher expression during the imbibition stages compared to the α2 and α3 isoforms, which experienced a drastic reduction in expression from 5 DAI. This contrasting pattern may be attributed to phylogenetic and functional differences among α-conglutin isoforms [51]. Furthermore, similar expression trends were observed in the γ-conglutin family, with γ1 and γ2 isoforms showing decreased expression as germination progressed. However, γ2 conglutin displayed higher expression during the early germination stages, but this decreased by 6 DAI compared to the γ1-conglutin isoform. Additionally, the 4δ-conglutin isoform exhibited a reduction in expression by 5 DAI, while δ1-conglutin decreased at 7 DAI. These disparities in α- and γ-conglutin expression dynamics highlight the intricate regulatory mechanisms governing conglutin gene expression during NLL seed germination [38].

Another noteworthy distinction was observed in the β-conglutin family, where our findings revealed diverse transcriptional patterns among isoforms. In contrast to the conventional trend of storage protein degradation during germination, most β-conglutin isoforms in our study exhibited decreased expression levels. However, the standout exceptions were β4 and β6 conglutins, which displayed increased expression levels at later germination stages. Notably, β4 surpassed the 5-day mark after imbibition (DAI), while β6 peaked around 3 DAI. These atypical expression patterns deviated from Foley et al.’s observations and underscore the unique functional and physiological roles that specific β-conglutin isoforms may play during the germination process [38].

While our study aligns with some aspects of Foley et al.’s data, such as the general trend of storage protein degradation, it also elucidates previously unrecognized complexities in conglutin gene regulation during NLL seed germination. These differences underscore the need for further investigation to fully unravel the functional significance of conglutins in the context of seed germination and seedling development.

### 3.4. Oxidative Enzyme Modifications during Seed Germination

Our analysis of the expression of oxidative stress genes related to the AsA-GSH cycle during lupine seed germination provided information about the various enzymes involved in this process, as shown in Figure 4. The AsA-GSH cycle enzymes are ascorbate peroxidase (APX), monodehydroascorbate reductase (MDHAR), dehydroascorbate reductase (DHAR), and glutathione reductase (GR) [52,53]. Apart from these enzymes, the products of the AsA-GSH cycle, which are ascorbate (Asc) and glutathione (GSH), can interact directly with ROS, such as superoxide and hydroxyl radicals, resulting in the non-enzymatic scavenging of oxidative species, thereby bolstering the antioxidative defense system [27,54]. Enzymatic expression showed that nitric oxide synthase (NOS), Glutathione synthetase (GS), Glutathione S-transferase (GsT), and γ-Glutamyl-cysteine synthase (γ-GCS) were more strongly expressed at early germination. This makes sense since nitric oxide (NO) has a role in dormancy alleviation, and therefore, its expression is crucial in dormancy breaking, supporting redox and energy balance in germinating seeds. Thus, the expression of NOS is a signal promoting seed germination. NO also nitrosylates proteins and glutathione, while ROS can interact with NO and facilitate its scavenging, leading to an interplay between them [18,55,56]. Moreover, recent research reports have shed light on NO’s multifaceted influence on antioxidant activity, with evidence suggesting its role in restraining the carbonylation of enzymes involved in the glutathione–ascorbate cycle, including GR, GST, APX, and DHAR [57]. This intricate orchestration of NO’s regulatory functions in tandem with the antioxidant defense machinery underscores its intricate involvement in modulating germination dynamics.

Glutathione synthetase (GS), Glutathione S-transferase (GST), and γ-Glutamyl-cysteine synthase (γ-GCS) emerge as pivotal GSH-associated enzymes that wield substantial influence within the AsA-GSH cycle, contributing to the reduction of H_2_O_2_ into water to uphold GSH levels [58,59,60,61]. This intricate process entails a two-step GSH biosynthesis pathway. The first step involves the conjugation of L-cysteine and L-glutamate, resulting in the formation of γ-Glutamyl-cysteine by γ-GCS. Subsequently, glycine is appended by GS, facilitated by the expenditure of two ATP molecules [62]. The maintenance of reduced GSH levels is overseen by glutathione reductase (GR), which executes amide bond formation between GSH and nicotinamide adenine dinucleotide phosphate (NADPH) [63]. Once synthesized in the cytosol, GSH is transported to other organelles as required for metabolic functions. However, under stress conditions, GSH’s rapid conversion into oxidized glutathione or glutathione disulfide (GSSG) can transpire through various biochemical reactions across different compartments. This dynamic equilibrium is orchestrated by the concerted actions of GR and GPX enzymes, which serve to regulate the delicate GSH/GSSG balance and mitigate adverse oxidative effects. Beyond its detoxification role against reactive oxygen species (ROS) such as H_2_O_2_, superoxide (O_2_^−^), and hydroxyl radical (HO·) through glutathionylation, GSH plays a multifaceted role in modulating the AsA-GSH cycle enzymes, further augmenting antioxidant defense mechanisms. Notably, studies have demonstrated that the application of glutathione induces elevated levels of abscisic acid (ABA), a crucial regulator of seed germination, stomatal aperture, and transpiration rate, thus underscoring GSH’s multifunctional involvement [64]. As expected, GR and GPX enzymes’ higher expression was observed during later stages of development because they maintain a balanced state of GSH/GSSG [65]. Similarly, our results showed higher gene expression in Cu/Zn superoxide dismutase (Cu/Zn-SOD), monodehydroascorbate reductase (MDHAR-chl), and catalase (CAT) enzymes at later stages of seed development (Figure 4). Particularly, the Cu/Zn-SOD enzyme has Cu (II) plus Zn (II) at the active site and is found in the cytosol and plastids. Its catalytic role in dismutating two superoxide radicals (O_2_^−^) and water into H_2_O_2_ and O_2_ is further complemented by CAT, ensuring efficient H_2_O_2_ scavenging. This concerted action of Cu/Zn-SOD and CAT establishes a robust front-line defense against oxidative stress, specifically within plastids, orchestrating an essential protective mechanism in plant cells [66].

The comprehensive analysis of conglutin expression across various stages of seed development presents intriguing insights into their roles as reserve proteins and potential mediators of oxidative stress. Considering our results and findings, all the conglutins showed decreasing expression throughout seed development, a pattern suggestive of their primary function as storage reserves (Figure 2 and Figure 3). Similarly, it can also be hypothesized that they may induce the activation of oxidative stress, as the expression of ROS enzymes is at a later DAI, which favors dormancy alleviation and subsequent seed germination. Several researchers have identified that antioxidative defense systems play important roles in the acquisition of seed desiccation tolerance [20,67,68,69]. However, the profile of β4 and β6 conglutin isoforms shows late expression at 5–6 and 3 DAI. These conglutins have been previously studied at the cellular level and can accumulate in the plasma membrane of plant cells where NADP is accumulated, the main ROS-producing enzyme [39]. Additionally, variations between conglutin isoforms could arise from nuanced physiological distinctions within seeds. Notably, the onset of imbibition involves a rapid influx of water into dry legume seeds, causing testa rupture and radicle protrusion. In related species like soybean, peroxidases emerge as major defense proteins during this imbibition phase.

In the larger context of our results, the remarkable diversity in both conglutin and antioxidant gene expression patterns fosters the intriguing hypothesis that specific conglutin isoforms might be implicated in orchestrating intermediary products of oxidative signaling pathways (Figure 3 and Figure 4). Concretely, the variations in different β- and γ-conglutin isoforms seems to be linked to important changes in some oxidative stress-related genes during NLL seed germination, which could be at the origin of redox signaling and seedling development. For example, as observed in both figures, the higher peaks of β4 at 5 DAI correspond to augmentations in GPx, GsT, MDHAR-cyt, and DHAR-cyt levels simultaneously, which indicates an enhanced oxidative stress response as an effort to counteract and neutralize ROS or other oxidative stressors that can lead to cellular damage [63]. The augmentation of the β4 isoform during seed germination could have the aim of maintaining redox balance and mitigating potential harm caused by excessive ROS. Following the same pattern, the augmentation of β6 at 3 DAI corresponds to a simultaneous peak of γ-GCS, GsT, and DHAR-Myt. As another example that supports our hypothesis, the peaks of β3 and γ2 at 2 DAI also correspond with rare features in the oxidative panel, simultaneously with the decrease of γ-GCS levels, reaching almost zero, meaning a disruption or reduction in the synthesis of γ-glutamyl cysteine [29]. This compound is a precursor in the biosynthesis of glutathione, a crucial antioxidant molecule in cells that plays a pivotal role in defending against oxidative stress and maintaining redox balance.

Considering the broader implications of our findings, the notable variation observed in the expression patterns of both conglutins and antioxidant genes gives rise to an intriguing proposition. This suggests that specific conglutin isoforms could potentially play a role in coordinating intermediary products within oxidative signaling pathways, with major implications in the redox control of seed germination and seedling development.

## 4. Conclusions and Future Perspectives

This study presents compelling insights into the dynamics of conglutin mobilization, their gene expression patterns, and ROS balance-dependent genes during NLL seed germination. These findings shed light on the intricate molecular events that underlie this critical phase of plant development. However, we acknowledge the need for further research to establish the functional connections between these components and the specific signaling roles of the β- and γ-conglutin families. To bridge this gap, future investigations will focus on experimental validation to confirm the functional significance of NLL conglutins as signaling molecules during germination progression. Moreover, we will explore the interplay between conglutins and hormonal regulation to gain a more comprehensive understanding of their interactive roles in seed germination.

Beyond the scope of germination, our research opens avenues for broader explorations into the impact of conglutin protein families on subsequent stages of plant development. Investigating their contributions to morphogenesis and developmental processes will provide a holistic view of their influence on the plant life cycle. Furthermore, delving into conglutin interactions within the plant and its environment will uncover their roles in stress responses and adaptive strategies, contributing to our understanding of plant resilience.

In conclusion, this study serves as a foundation for future research aimed at substantiating the functional significance of NLL conglutin proteins, particularly those from the β- and γ-families, in germination, and unraveling their broader implications for plant growth, development, and adaptability.

## Figures and Tables

**Figure 1 genes-14-01889-f001:**
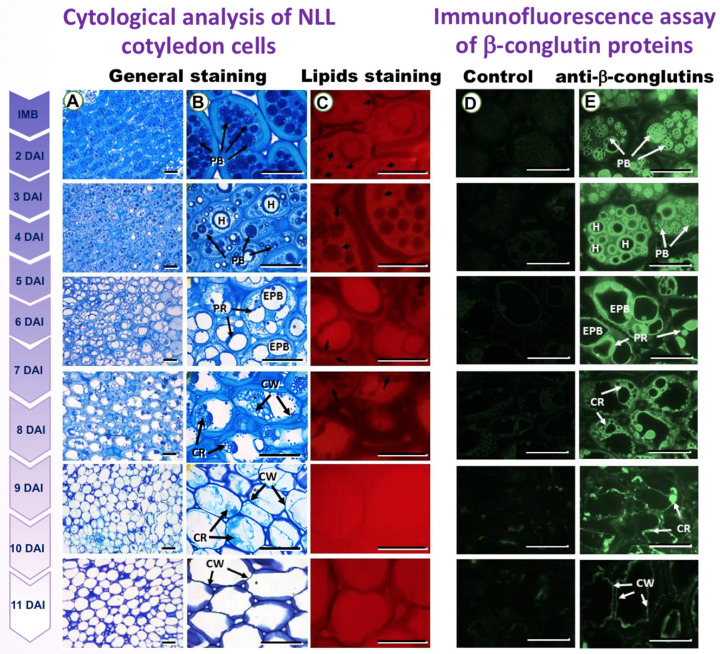
Microscopy study of NLL cotyledon during seed germination. (**A**,**B**) Microscopy images obtained from semithin 1μm sections of NLL cotyledon samples from different germination stages (such as IMB, 2, 5, 7, 9, and 11 DAI, respectively), after a general staining using 0.05% (*w*/*v*) methylene blue and 0.05% (*w*/*v*) toluidine blue solution. At IMB stage, PBs are full of storage proteins. Thenceforth, PBs empty their contents, leaving holes (H) through degradation of proteins used as nutrients for plant germination and growth, forming protein (PR) remains located at the PBs periphery. Holes increase their volume, creating empty protein bodies (EPB) that fuse themselves until complete disappearance. At this stage, cellular remains (CR) can be observed inside the cells. About 9–11 DAI, only the cell wall (CW) is present. (**C**) Microscopy images of 1μm lupin cotyledon sections stained via lipid-specific Nile Red staining. Arrowhead = lipid drop. (**E**) Immunocytochemical localization of β-conglutins and (**D**) control assay. It is demonstrated that β-conglutins co-localize with PBs and the remains they form after their degradation. CR: cellular remains; CW: cell wall; EPB: empty protein bodies; H: hole; PB: protein body; PR: periphery protein bodies’ remains. Scale bar: A panel: 100 µm; B–E panels: 50 µm.

**Figure 2 genes-14-01889-f002:**
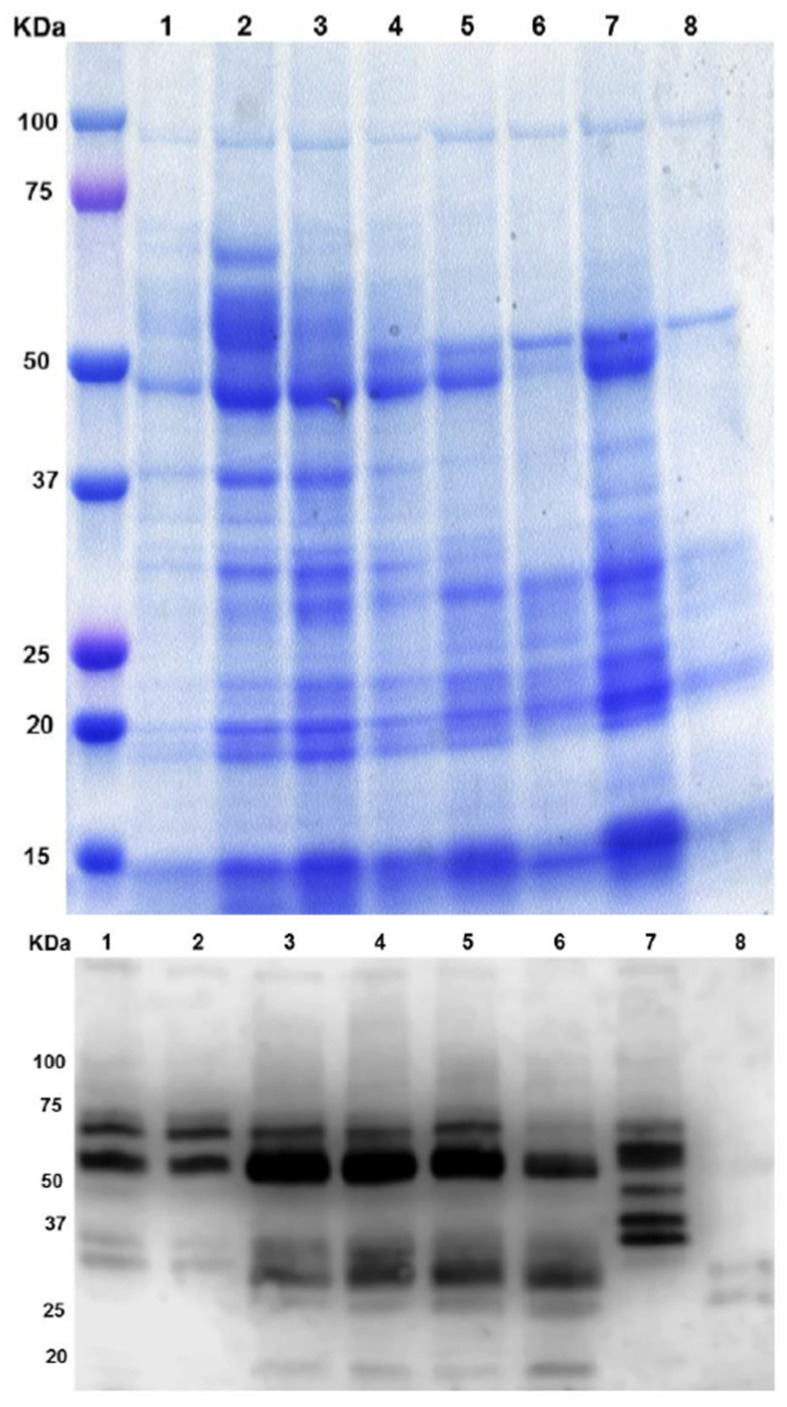
Biochemical analysis of β-conglutin proteins during NLL seed germination process. Upper panel: protein profile obtained via SDS-PAGE; lower panel: immunoblotting using a specific antibody against β-conglutins. 1: day of imbibition (IMB), 2–8: subsequent days after imbibition (DAI) stages—2, 3, 4, 5, 7, 9, and 11, respectively, designated as numbers 2 to 8). The primary antibody used was an anti-β-conglutin antibody developed in-house (custom-made, Agrisera) following Jimenez-Lopez et al. [39], a rabbit polyclonal anti-serum.

**Figure 3 genes-14-01889-f003:**
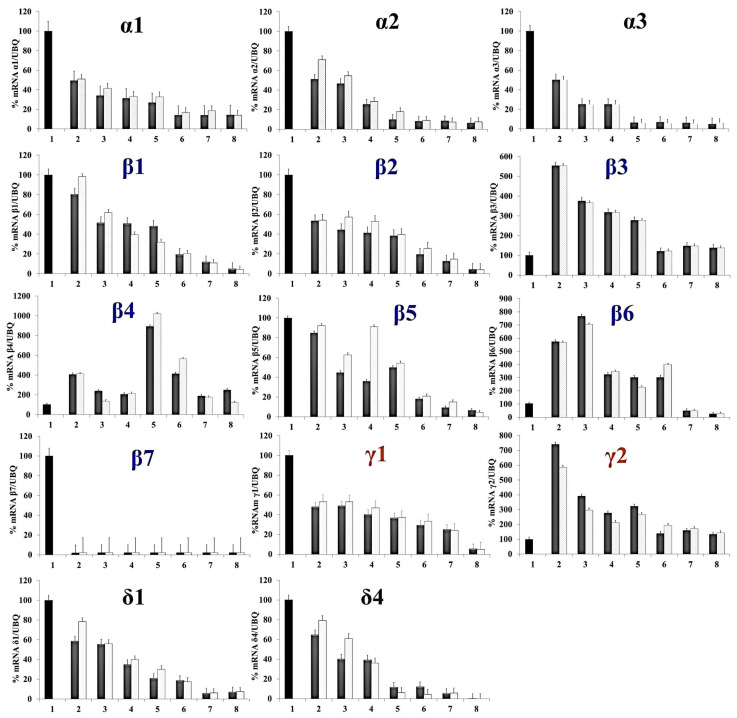
Expression analysis of conglutin family α, β, γ, and δ genes during seed germination. The housekeeping gene used as a control was ubiquitin (UBQ). Numbers on X-axis represent days after imbibition (DAI, numbers 2–8). Black bars: imbibition (number 1) and grey bars: seeds germination without using MS media. White/light greys bars: seeds germinated in MS medium. In all of the graphs, all grey and white/light grey bars (corresponding to the expression of the different genes at different stages post-imbibition with and without using MS media) exhibit statistically significant differences (*p* < 0.05) compared to their respective imbibition stages (black bars, stage 1).

**Figure 4 genes-14-01889-f004:**
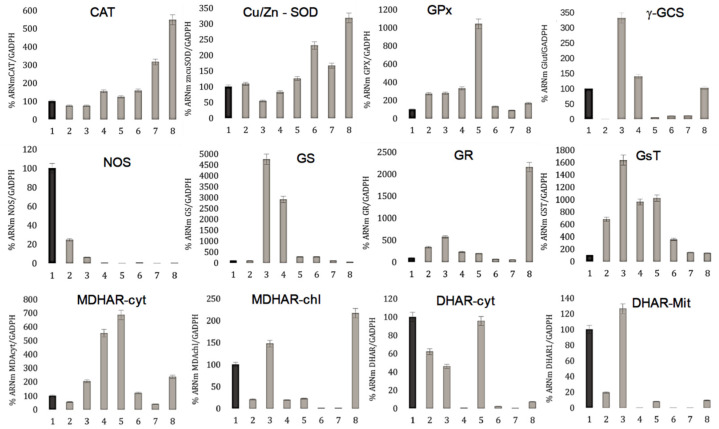
Expression of oxidative stress-related genes during NLL seed germination. The housekeeping gene used as a control was Glyceraldehyde-3-phosphate dehydrogenase (GADPH). Genes analyzed correspond to the AsA-GSH cycle as the head of the antioxidant enzyme system, such as GPx: Glutathione peroxidase, γ-GCs: γ-Glutamyl-cysteine synthase, GS: Glutathione synthetase, GR: Glutathione reductase, GsT: Glutathione S-transferase; MDHAR: Monodehydro-ascorbate reductase, and DHAR: Dehydro-ascorbate reductase, as well as NOS: Nitric oxide synthase, CAT: catalase, Cu/Zn-SOD: Cu/Zn superoxide dismutase, cyt: cytoplasm, chl: chloroplasts, and mit: mitochondria. In all cases (measured enzymes) and at every stage, the values exhibit significant differences (*p* < 0.05) compared to the black bar, representing the imbibition state, except for Cu/Zn-SOD at stage 2, GS at stages 2 and 7, and DHAR cyt at stage 5, where no statistically significant differences were found.

**Table 1 genes-14-01889-t001:** Primers for the analysis of gene expression related to oxidative metabolism.

Primer	Forward	Reverse
CAT	GGA ACT ATC CCG AGT GGA AAC	CCT CAG GCC AAG TCT TAGT TAC
GsT	GGA CCC AAA TGA TGG AAC AGA	GCC AAA CCC AAG TCA ACA AC
GS	CAC TAC CAC CAC CAC TCA AA	TGC GAG GTT CAC GGA TTT
GPx	CAA GGA TGA TGC GGA GTA TGT	TGA AAC CTC CTG TGC CAT AAA
GR	GGA GCC AAG GTT GGG ATT T	GGG AAC ACA GCC ACG AAT AA
Cu/Zn-SOD	GGG TCA CCT GGG AAA CAT AG	CCA CTA AGG CTC TTC CAA CTA C
NOS	CCA GAG GTT TGC CTC AGA TT	TTC ACC AGA TGA ACG GAT TG
MDHAR-chl	GAA ACC TAT CCG GTG TTC ACT AT	TTC ACC TCC AAC AAC TAC AAC
MDHAR_cyt	GGA CGA GAG CAG ATT TCC ATA A	AAA GGA GAA GGG AAA GTG TGA G
DHAR-chl	CTC CTC CTT CCC AAC CAT TT	TTC CTC CAG TGT CAG CAA TAC
DHAR_cyt	GAG ACA AGG CTG AGG GTA TTT	GGA GAT AAG TCC AGA AGG GAA AG
GAPDH	CGT GTC CCT ACA GTT GAT GTT	CCT CCT TGA TGG CAG CTT TA

**Table 2 genes-14-01889-t002:** Primers for the analysis of gene expression of conglutins.

Primer	Forward	Reverse
α1	GAA CAA GGG CAA GGT CCT A	CAC CAG TAG GCA CTG CTA
α2	GAA GAA ACA AGT ACT CGA G	GTT CTT GRC CTT GTT CTT G
α3	CAC GGC GAA GAC ATA CAC G	CTC FFT FFF FAT TFT TCC T
β1	AGG AAT ACG AGC AAG GAG AGG	GAA AGT GGA TGA CCT GCC G
β2	AAG AAG GGG TTT ATC TTT TCC	CAT GAT AGT AGA TAG CAG CTT AG
β3	GGA ACA GAG GGA AGA GAG AGA AC	CAT TGG CTC GTC ATC TTC CTG C
β4	CAG AAG TTA AAG GGC TCA	TTG TTG TTG TTG GGG T
β5	CAG AGT GGT GAT GAG AGA AAG AC	CAT TGT CTG CCT GAG CT
β6	GAG CGT GAT CGT GAG CCT T	GTT CTC TCT CTT CCC TCT GTT CC
β7	GTG AGA TTT CCA ACC TTA GTC	TTG TTT ACG CGG ACG TG
γ1	ATG GCA CGA AAT ATG GCT CA	AGA AGT GGG TAG AGG ATA ATG C

## Data Availability

Data Sharing is not applicable to this article.

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
