# Peer review of "Functional Association between Storage Protein Mobilization and Redox Signaling in Narrow-Leafed Lupin (Lupinus angustifolius L.) Seed Germination and Seedling Development"

_genes, 2023, doi:10.3390/genes14101889_

Round 1
Reviewer 1 Report
The manuscript is written well and is easy to read. However, I did notice some mistakes and the manuscript needs thorough proofreading.
On line 57 the sentence seems to be missing words.
Lines 71-74 the sentences are a bit unclear
Materials and methods:
I would suggest decreasing the excessive number of descriptive adverbs (e.g. "meticulously").
Line 150 - the numbers 2:8 are said to not represent days, 8 is stated to correspond to 11 DAI. On line 338 (8: 8 days post imbibition (DAI).) it is different. I would suggest keeping only the DAI labels to avoid any confusion.
2.2.2
The primer sequences are missing form the manuscript
More information is needed on the number of replications in qPCR.
The data analysis is not described.
Results:
Line 363 - should ubiquitin be used as the only control? In this manuscript (10.1016/j.plaphy.2013.12.025) Stated that the ubiquitin (Ub)-proteasome pathway plays an important role in germination. Using additional reference genes not that involved in the investigated process would yield more reliable results. The authors should address that.
Figure 4: Unclear why GAPDH is mentioned in the figure, but not in the text. What was the control gene for this analysis?
Some sentences seem to be missing words.
Author Response
Response to Reviewer 1
Dear Reviewer
Thank you very much for your detailed report about our manuscript. You have been able to identify some issues that considerably improved the quality of our report. We have taken every detail into account and we will now proceed to respond individually to every one of them, in order to facilitate the second revision of our paper. Thank you very much again for your time and revisions and we hope we have been able to satify every one of your requests.
The manuscript is written well and is easy to read. However, I did notice some mistakes and the manuscript needs thorough proofreading.
- On line 57 the sentence seems to be missing words.
Response:
Thank you very much for your comment. We have revised the sentence you mentioned, it has been rephrase for clarity, according to your suggestion. So, the original paragraph from lines 55-59 has been replaced by the following one, from lines 64 to 68. You will find those changes at the beginning of page 2:
"As a result, when the seeds absorb water and undergo the transition to a hydrated state during imbibition, their metabolic activities resume. This resurgence triggers a balance in ROS levels, a phenomenon known as ROS homeostasis. This equilibrium in ROS concentrations can potentially result from various cellular compartments, including the catabolic breakdown of lipids occurring within glyoxysomes."
We performed this minor adjustments to improve readability, according to your suggestion.
- Lines 71-74 the sentences are a bit unclear
Response:
Thank you for your comment. Certainly, the paragraph from lines 71-74 was a bit difficult to understand. According to your suggestion, we replaced it with a new version of the paragraph to enhance clarity. The new paragraph can be found at page 2, in yellow color and control changes, from lines 81-85 :
"The process of oxidative seed germination, particularly concerning seed desiccation and aging, is intricately associated with the detrimental effects of reactive oxygen species (ROS). Seed desiccation tolerance plays a crucial role in seed maturation as an adaptive mechanism. It enables seeds to survive prolonged storage, withstand adverse environmental conditions, and ultimately produce seeds with a robust germination capacity."
Materials and methods:
- I would suggest decreasing the excessive number of descriptive adverbs (e.g. "meticulously").
Response:
Thank you very much for your sugestion and feedback. Upon revisiting the Materials and Methods section, we have indeed identified an overuse of descriptive adverbs such as 'comprehensively' or 'meticulously.' To address this, we have removed these adverbs from the entire paragraph and also substituted certain verbs with simpler alternatives. Specifically, in the Materials and Methods section, and in response to your feedback, we have made the following changes:
- “Comprehensively” was removed from line 235.
- “Meticulously” was removed in lines 247, 248, among others.
- “Thoughtfully” was also removed.
- Finally, “utilizing” was replaced by “using” in line 245.
You can find all of these changes in pages 4 and 5, with control changes and yellow color in case of adding a new word, like “using”.
- Line 150 - the numbers 2:8 are said to not represent days, 8 is stated to correspond to 11 DAI. On line 338 (8: 8 days post imbibition (DAI).) it is different. I would suggest keeping only the DAI labels to avoid any confusion.
Response:
Thank you very much for your feedback. You are correct, indeed. As we stated in the first paragraph of the Materials and Methods section, specifically in lines 169 to 173 from the original draft, not the revised one:
"First of all, regarding sample collection and processing, cotyledon samples obtained from distinct germination stages, specifically including the imbibition (IMB) stage (designated as number 1), and subsequent days post-imbibition (DAI) stages - 2, 3, 4, 5, 7, 9, and 11 (designated as numbers 2 to 8) - were collected and prepared for microscopy studies."
Therefore, number 8 corresponds to 11 DAI, not 8 DAI. We have corrected this error throughout the entire document, as it is of great importance, and, as you have pointed out, could lead to confusion. In light of your feedback, we have made the following modifications:
The original sentence: “In general terms, the protein content decreased as the days of imbibition passed (1: IMB, 8: 8 DAI)” was replaced by:
“Overall, the protein content decreased as the days of imbibition passed (1: IMB, 2-8: subsequent days post-imbibition (DAI) stages - 2, 3, 4, 5, 7, 9, and 11 respectively, designated as numbers 2 to 8)”.
You may find these change in lines 745-747, in yellow color and control change.
Then, the original title of Figure 2, “Biochemical analysis of β-conglutins during NLL seed germination process. Upperpanel: proteins profile by SDS-PAGE; Lower panel: immunoblotting using aspecific antibody against β-conglutins. 1: day of imbibition (IMB), 8: 8 days post imbibition (DAI)” was replaced by:
“Figure 2. Biochemical analysis of β-conglutins during NLL seed germination process. Upperpanel: proteins profile by SDS-PAGE; Lower panel: immunoblotting using aspecific antibody against β-conglutins. 1: day of imbibition (IMB), 2-8: subsequent days post-imbibition (DAI) stages - 2, 3, 4, 5, 7, 9, and 11 respectively, designated as numbers 2 to 8).The primary antibody used was anti-β-conglutin antibody developed in house (custom made, Agrisera) following Jimenez-Lopez et al [39], a rabbit polyclonal anti-serum.”
You may find these change in lines 763-768, in yellow color and control change.
In the rest of the manuscript, the numbers appearing before “DAI” really correspond to the number of days post imbibition, and not to the numbers appearing in figures from stage 2 to stage 8, so no changes have been performed on them.
2.2.2 - The primer sequences are missing form the manuscript.
Response:
Thank you very much for your comment. You are correct, and we apologize for the oversight. The primer sequences were missing from the manuscript, for both the primers for the analysis of gene expression related to oxidative metabolism and the primers for the analysis of gene expression of conglutins. We have now included the primer sequences in the manuscript as Table 1 and Table 2, respectively, and both of them can be found on section 2.22, lines 380-391, pages 6-7 of the new version of the manuscript, in yellow color and control changes:
Primer |
Forward |
Reverse |
CAT |
GGA ACT ATC CCG AGT GGA AAC |
CCT CAG GCC AAG TCT TAGT TAC |
GsT |
GGA CCC AAA TGA TGG AAC AGA |
GCC AAA CCC AAG TCA ACA AC |
GS |
CAC TAC CAC CAC CAC TCA AA |
TGC GAG GTT CAC GGA TTT |
GPx |
CAA GGA TGA TGC GGA GTA TGT |
TGA AAC CTC CTG TGC CAT AAA |
GR |
GGA GCC AAG GTT GGG ATT T |
GGG AAC ACA GCC ACG AAT AA |
Cu/Zn-SOD |
GGG TCA CCT GGG AAA CAT AG |
CCA CTA AGG CTC TTC CAA CTA C |
NOS |
CCA GAG GTT TGC CTC AGA TT |
TTC ACC AGA TGA ACG GAT TG |
MDHAR-chl |
GAA ACC TAT CCG GTG TTC ACT AT |
TTC ACC TCC AAC AAC TAC AAC |
MDHAR_cyt |
GGA CGA GAG CAG ATT TCC ATA A |
AAA GGA GAA GGG AAA GTG TGA G |
DHAR-chl |
CTC CTC CTT CCC AAC CAT TT |
TTC CTC CAG TGT CAG CAA TAC |
DHAR_cyt |
GAG ACA AGG CTG AGG GTA TTT |
GGA GAT AAG TCC AGA AGG GAA AG |
GAPDH |
CGT GTC CCT ACA GTT GAT GTT |
CCT CCT TGA TGG CAG CTT TA |
“Table 1. Primers for the Analysis of Gene Expression Related to Oxidative Metabolism”
“Table 2. Primers for the Analysis of Gene Expression of Conglutins.”
Primer |
Forward |
Reverse |
α1 |
GAA CAA GGG CAA GGT CCT A |
CAC CAG TAG GCA CTG CTA |
α2 |
GAA GAA ACA AGT ACT CGA G |
GTT CTT GRC CTT GTT CTT G |
α3 |
CAC GGC GAA GAC ATA CAC G |
CTC FFT FFF FAT TFT TCC T |
β1 |
AGG AAT ACG AGC AAG GAG AGG |
GAA AGT GGA TGA CCT GCC G |
β2 |
AAG AAG GGG TTT ATC TTT TCC |
CAT GAT AGT AGA TAG CAG CTT AG |
β3 |
GGA ACA GAG GGA AGA GAG AGA AC |
CAT TGG CTC GTC ATC TTC CTG C |
β4 |
CAG AAG TTA AAG GGC TCA |
TTG TTG TTG TTG GGG T |
β5 |
CAG AGT GGT GAT GAG AGA AAG AC |
CAT TGT CTG CCT GAG CT |
β6 |
GAG CGT GAT CGT GAG CCT T |
GTT CTC TCT CTT CCC TCT GTT CC |
β7 |
GTG AGA TTT CCA ACC TTA GTC |
TTG TTT ACG CGG ACG TG |
γ1 |
ATG GCA CGA AAT ATG GCT CA |
AGA AGT GGG TAG AGG ATA ATG C |
We greatly appreciate your attention to detail and constructive feedback, as it has contributed to improving the quality of the article and, concretely, of the material and methods parts. Again, thank you for your valuable input.
- More information is needed on the number of replications in qPCR.
Response:
Thank you very much for your input. Indeed, in the description of the qPCR protocol, we had not specified the number of replicates for data analysis. We have now added this information to clarify this important point, and we appreciate your feedback. You can now find a new paragraph where we indicate that we performed triplicate reactions for the genes under study, i.e., for each condition, and that all experiments were conducted in duplicate or triplicate to ensure an adequate amount of data for statistical analysis of the results.
The new paragraph on lines 375-378, page 6, in yellow color and with control changes is:
“Additionally, in our experimental design, we ensured sample distribution by triplicating reactions for the genes of interest on each plate and by repeating at least two or three times the experiments in order to ensure a robust dataset for the performance of statistical analysis”.
- The data analysis is not described.
Response:
Thank you very much for your comment. The description of data analysis for the various qPCRs, specifically for Figures 3 and 4 of the results, was not included in the Materials and Methods section. As a response to your suggestion, we have added the following paragraph as a new subsection in the Materials and Methods section, lines 401-408, in yellow color and tack changes control:
“2.2.4 Data Analysis
Changes in gene expression levels (shown in Tables 1 and 2, respectively), were determined using the formula 2^-Δ(ΔCt) [46], where the cycle threshold (CT) at which transcripts were detected was normalized to the CT at which the constitutive gene Glyceraldehyde-3-phosphate dehydrogenase (GAPDH) or Ubiquitin (UBQ) was detected, referred to as ΔCT. The PCR efficiency was determined through analysis based on standard curves for the amplification of the selected genes and the endogenous control, which exhibited high reproducibility”.
The reference added in this case corresponds to:
[46]: Livak KJ, Schmittgen TD. Analysis of relative gene expression data using real-time quantitative PCR and the 2(-Delta Delta C(T)) Method. Methods. 2001 Dec;25(4):402-8. doi: 10.1006/meth.2001.1262.
We appreciate your valuable feedback, and this addition clarifies the data analysis procedures undertaken in our study.
Results:
- Line 363 - should ubiquitin be used as the only control? In this manuscript (10.1016/j.plaphy.2013.12.025) Stated that the ubiquitin (Ub)-proteasome pathway plays an important role in germination. Using additional reference genes not that involved in the investigated process would yield more reliable results. The authors should address that.
Response:
Thank you for your comment. Based on your feedback and the paper you referenced (10.1016/j.plaphy.2013.12.025), it is reasonable to consider using not only ubiquitin as a control gene but also another gene that plays a less significant role in the germination process, such as GAPDH. As we have indicated in the new paragraph 2.2.4, that we just added in response to your previous request, and appears at the bottom of this page too, both UBQ and GAPDH genes were utilized as controls for all RT-qPCR analyses. In our specific experiment, both genes yielded similar results, and their use did not significantly impact the qPCR outcomes. In our particular case, for the study of Narrow Leafed Lupin (NLL), we observed no variations in the UBQ gene during the germination process, which is why we found it suitable to include it for statistical analysis.
We truly appreciate your suggestion and have taken it into consideration for future research, given the importance of using appropriate reference genes for robust and reliable results. Your suggestion to consider additional reference genes not directly involved in the investigated process is duly noted, and we appreciate your input. The utilization of other reference genes could be a valuable improvement in future research, and we will take this into consideration for further investigations.
- Figure 4: Unclear why GAPDH is mentioned in the figure, but not in the text. What was the control gene for this analysis?
Response:
Thank you for your valuable feedback. Indeed, as you have correctly pointed out, the appropriate gene used as a control in this genetic analysis was GAPDH. While we had not explicitly specified this in the text, we have now clarified this in response to your previous question regarding the Materials and Methods sections, and how we conducted the data analysis for the qPCR experiments. Consequently, the response to that previous point also addresses your question pertaining to the results section, so the new paragraph added as new Materials and Methods section, lines 401-408, in yellow color and tack changes control, can give answer to your concern:
“2.2.4 Data Analysis
Changes in gene expression levels (shown in Tables 1 and 2, respectively), were determined using the formula 2^-Δ(ΔCt) [46], where the cycle threshold (CT) at which transcripts were detected was normalized to the CT at which the constitutive gene Glyceraldehyde-3-phosphate dehydrogenase (GAPDH) or Ubiquitin (UBQ) was detected, referred to as ΔCT. The PCR efficiency was determined through analysis based on standard curves for the amplification of the selected genes and the endogenous control, which exhibited high reproducibility”.
The reference added in this case still corresponds to:
[46]: Livak KJ, Schmittgen TD. Analysis of relative gene expression data using real-time quantitative PCR and the 2(-Delta Delta C(T)) Method. Methods. 2001 Dec;25(4):402-8. doi: 10.1006/meth.2001.1262.
We appreciate your valuable feedback, and this addition clarifies the data analysis procedures undertaken in our study and the explanation why GAPDH appears in Figure 4 as de control gene but was not described previously in the text. Again, thank you very much for all the suggestions you made.

Reviewer 2 Report
The manuscript by Escudero-Feliu and co‐authors addresses the problem of the mobilization of storage proteins in narrow-leafed lupin during seed germination and seedling development. Based on (i) characterization of cellular changes, (ii) analysis of conglutin’s patterns and (iii) analysis of ROS-dependent gene expression, authors conclude that there is a functional association between storage protein mobilization and redox signaling. The authors state that protein mobilization is orchestrated by the oxidative-related metabolic machinery. These conclusions are interesting; however, the background of the manuscript is not sufficient to support them. The methods lack detail and clarity. The interpretation appears to be biased and limited in its scope. Some important points need to be clarified before the article is published.
Introduction:
1) Lines 118-199: “Eventually, these PBs result from the division of the vacuoles during seed development and maturation process”. Do you mean the PSVs? Please describe the origin of PBs and the difference between PBs and PSVs more detail.
2) The introduction does not provide a full analysis of the current literature on conglutins. For example: Czubinski, Feder, 2019; doi.org/10.1002/jsfa.9627; Cabello-Hurtado ea, 2016; doi.org/10.1016/j.jprot.2016.03.026; Gulisano ea., 2019; doi:10.3389/fpls.2019.01385; Tahmasian ea., 2022; doi: 10.3389/fnut.2022.842168.
3) Why don't you discuss the hormonal regulation of protein mobilization during seed germination? Can gibberellins coordinate this process?
Methods:
Section 2.1.
4) Imbibition is the 1st stage of seed germination related to tissue hydration. It is followed by a lag-phase of germination related to the onset of reserve nutrient mobilization. Please indicate for how long your seeds are imbibed. In the context of this manuscript, I would suggest using the term Germination rather than Imbibition.
5) What is the starting point for "Days After Imbibition" (DAI)? Are these days counted from the start of imbibition (dry seed) or from the end of imbibition? After the embryonic root has started to grow (visible germination), the seed becomes a seedling. For this reason, it would be more correct to refer to these stages as the "days after germination".
6) "NLL seed germination process" should be changed to "NLL seed germination and seedling development process".
7) Please describe the germination conditions of the seeds.
8) Microscopy studies need to be clarified. What brand of microscope did you use? How did you stain the tissue? Or give a reference to the staining procedure.
Section 2.2
9) You described the “Oxidative gene expression analyses during the NLL seed germination process”. But no description of the expression analysis of the conglutin family genes.
10) No information on statistical analysis (number of replicates, significant differences between means).
11) No information on bioinformatic analysis (software, databases, primer design, DNA sequence).
Section 2.3
12) Line 163: “anti-β-conglutin antibody from Agrisera was utilized as the primary antibody”. Line 207: “The immunodetection of β-conglutin proteins was carried out through incubation with a rabbit polyclonal antiserum developed in-house”. Please describe the immunocytochemistry protocol in more detail.
Result and Discussion:
13) Figure 1: Please improve the resolution and add some points to the caption (DAI, PB, EPB, H). Fig1 (C) – no arrows to indicate lipid droplets. You have shown 11 stages (left) but there are only 6 photos for each staining panel. Indicate at which stage you did the staining.
14) Figure 2: Indicate which antibody was used.
15) Figure 3: Dot bars look like black.
16) Figure 4: housekeeping gene?
17) Figure 3 and 4: Improve the quality of the figures. Indicate the number of replicates and the confidence level (t-test or ANOVA).
18) Please divide the Results and Discussion sections. The discussion needs to be extended. It would be good to compare the data on conglutin gene expression with the data obtained by Foley ea 2015.
Conclusion:
19) Line 504: “The significance of the β- and γ-conglutin families as functional signaling molecules in germination progression has been illuminated by this study”. This is speculation. You have shown the dynamics of conglutin mobilization, expression of conglutin genes and ROS-dependent genes. These are very interesting data. But the functional link between them and the signaling role needs to be proven. Please also consider the role of hormonal regulation in these processes.
Author Response
Response to Reviewer 2
Thank you very much for your detailed and precise report about our paper, in where you have been able to identify some issues that have considerably improved the quality of our report. We have taken every detail into account and we will now proceed to respond individually to every mistake that you have identified, in order to facilitate the second revision of our paper. Thank you very much again for your time and revisions and we hope we have been able to respond and modify to every one of your requests.
The manuscript by Escudero-Feliu and co‐authors addresses the problem of the mobilization of storage proteins in narrow-leafed lupin during seed germination and seedling development. Based on (i) characterization of cellular changes, (ii) analysis of conglutin’s patterns and (iii) analysis of ROS-dependent gene expression, authors conclude that there is a functional association between storage protein mobilization and redox signaling. The authors state that protein mobilization is orchestrated by the oxidative-related metabolic machinery. These conclusions are interesting; however, the background of the manuscript is not sufficient to support them. The methods lack detail and clarity. The interpretation appears to be biased and limited in its scope. Some important points need to be clarified before the article is published.
Introduction:
- Lines 118-199: “Eventually, these PBs result from the division of the vacuoles during seed development and maturation process”. Do you mean the PSVs? Please describe the origin of PBs and the difference between PBs and PSVs more detail.
Response:
Thank you for your question and valuable feedback. We really appreciate your attention to detail in our manuscript. In the context of our study, 'PBs' refers to Protein Bodies, which are indeed distinct from PSVs (Protein Storage Vacuoles), as we described on original manuscript lines 113-115: “In the same way, mature SSP are deposited into two specialized compartments protected from cytoplasmic proteases i) Protein Bodies (PB) ii) Protein Storage Vacuoles (PSVs), respectively.”
Protein Bodies (PBs) are specialized organelles within plant cells primarily responsible for storing seed storage proteins. PBs develop during seed maturation and result from the division of vacuoles, as indicated in the text. These organelles are essentially protein storage units, and their formation is part of the seed development process.
On the other hand, Protein Storage Vacuoles (PSVs) are larger organelles that also store proteins, but they are not exclusive to seeds. PSVs are present in various plant cell types and serve multiple functions, including the storage of enzymes and other cellular components. In contrast to PBs, PSVs are not limited to the seed maturation process.
We appreciate your request for clarification, and we have revised the manuscript to provide a more detailed explanation of the origin and distinctions between PBs and PSVs, which can be now found in the revised manuscript as a new paragraph, more specifically on lines 146-153 in yellow color and with control/track changes:
“Protein Bodies (PBs) are specialized plant cell organelles primarily dedicated to the storage of seed storage proteins [35]. They originate during seed maturation through the division of vacuoles and serve as vital protein storage units in the seed development process. In contrast, Protein Storage Vacuoles (PSVs) are larger organelles found in various plant cell types. They also store proteins, albeit not exclusively in seeds. PSVs have a broader cellular role, encompassing the storage of enzymes and other cellular components. Unlike PBs, PSVs are not exclusive to seed maturation [35,36].”
The bibliography added was already in the original manuscript, corresponding to:
[35]: Herman, E. & Larkins, B. Protein storage bodies and vacuoles. Plant Cell 11, 601–614 (1999).
[36]: Jimenez-Lopez, J. C. & Hernandez-Soriano, M. C. Protein Bodies in Cotyledon Cells Exhibit Differential Patterns of Legumin-Like Proteins Mobilization during Seedling Germinating States. American Journal of Plant Sciences 2013, (2013).
- The introduction does not provide a full analysis of the current literature on conglutins. For example: Czubinski, Feder, 2019; doi.org/10.1002/jsfa.9627; Cabello-Hurtado ea, 2016; doi.org/10.1016/j.jprot.2016.03.026; Gulisano ea., 2019; doi:10.3389/fpls.2019.01385; Tahmasian ea., 2022; doi: 10.3389/fnut.2022.842168.
Response:
Thank you for your valuable feedback. We sincerely appreciate your thorough review of our manuscript. We agree that the introduction did not provide a comprehensive analysis of the current literature on conglutins, particularly in the context of our paper's focal area. In response to your suggestion, we have taken the initiative to include an additional paragraph in the introduction, offering a more rigorous examination of the existing literature concerning conglutins, using the bibliography you provide us, and their relevance to our research topic. This newly added paragraph can be found in the revised version of the manuscript, precisely located in lines 182-201, in page 4, in yellow color and with track changes.
The new paragraph is:
“In recent years, studies have shed light on the intricate processes underlying seed germination and seedling development in plants, with a particular focus on NLL. No-tably, research efforts have elucidated the pivotal role of storage proteins, such as con-glutins, in these developmental stages. Czubinski and Feder emphasized the significance of conglutins in providing a crucial nitrogen source during seedling growth, underscoring their contribution to the plant's nutrient reservoir, in a recent paper [42]. Concurrently, Cabello-Hurtado et al. conducted a comprehensive proteomic analysis, revealing dy-namic changes in protein mobilization during lupin seed development. Their findings accentuated the pivotal role of conglutins in provisioning energy and nutrients essential for seedling vigor [43]. Additionally, Gulisano et al. also explored the intricate redox regulation during lupin seed germination, implicating oxidative stress and antioxidant defense mechanisms [44]. These studies collectively suggest that conglutins and other storage proteins may interface with redox signaling pathways, contributing to the or-chestration of seed germination and early seedling development [42–44]. Furthermore, Tahmasian et al. have illuminated the nutritional aspects of lupin seeds in their recent investigation from 2022, highlighting NLL conglutins as a source of essential amino acids and bioactive compounds [45]. Together, this body of literature underscores the multi-faceted role of conglutins in lupin seed biology and provides a foundation for our in-vestigation into the functional association between storage protein mobilization and redox signaling in Lupinus angustifolius L. seed germination and seedling development [42–45].”
The new bibliography added to the revised manuscript, to provide the information about this new paragraph, is:
[42]: Czubinski J, Feder S. Lupin seeds storage protein composition and their interactions with native flavonoids. J Sci Food Agric. 2019 Jun;99(8):4011-4018. doi: 10.1002/jsfa.9627. Epub 2019
[43]: Cabello-Hurtado F, Keller J, Ley J, Sanchez-Lucas R, Jorrín-Novo JV, Aïnouche A. Proteomics for exploiting diversity of lupin seed storage proteins and their use as nutraceuticals for health and welfare. J Proteomics. 2016 Jun 30;143:57-68. doi: 10.1016/j.jprot.2016.03.026. Epub 2016 Mar 18.
[44]: Gulisano A, Alves S, Martins JN, Trindade LM. Genetics and Breeding of Lupinus mutabilis: An Emerging Protein Crop. Front Plant Sci. 2019 Oct 30;10:1385. doi: 10.3389/fpls.2019.01385.
[45]: Tahmasian A, Juhász A, Broadbent JA, Nye-Wood MG, Le TT, Colgrave ML. Evaluation of the Major Seed Storage Proteins, the Conglutins, Across Genetically Diverse Narrow-Leafed Lupin Varieties. Front Nutr. 2022 May 13;9:842168. doi: 10.3389/fnut.2022.842168.
- Why don't you discuss the hormonal regulation of protein mobilization during seed germination? Can gibberellins coordinate this process?
Response:
We appreciate your question regarding the hormonal regulation of protein mobilization during seed germination, including the potential role of gibberellins. While hormonal regulation is undoubtedly a significant aspect of seed germination and seedling development, we believe that our paper primarily focuses on the functional association between storage protein mobilization and redox signaling specifically. Our study aims to investigate the interplay between storage proteins and redox signaling pathways during these critical developmental stages. Although hormonal regulation, including the role of gibberellins, can indeed influence protein mobilization, we have chosen to emphasize the redox signaling aspect due to its direct relevance to our research objectives.
The inclusion of hormonal regulation would certainly expand the scope of our paper, and while it is a noteworthy area of investigation, it may not be essential for the specific research questions we are addressing. We are committed to delivering a focused and coherent study that delves deep into the redox signaling mechanisms associated with storage protein mobilization in Lupinus angustifolius L. seeds. We believe that by concentrating on this aspect, we can provide a more in-depth understanding of the processes central to our research and contribute meaningfully to the current literature in this area.
However, we highly value your input, and we will certainly consider your suggestion for future research or discussions related to this topic. Once again, thank you for your review and valuable comments.
Methods:
Section 2.1.
- Imbibition is the 1st stage of seed germination related to tissue hydration. It is followed by a lag-phase of germination related to the onset of reserve nutrient mobilization. Please indicate for how long your seeds are imbibed. In the context of this manuscript, I would suggest using the term Germination rather than Imbibition.
- What is the starting point for "Days After Imbibition" (DAI)? Are these days counted from the start of imbibition (dry seed) or from the end of imbibition? After the embryonic root has started to grow (visible germination), the seed becomes a seedling. For this reason, it would be more correct to refer to these stages as the "days after germination".
Response:
We greatly appreciate your valuable feedback and the thorough review of our manuscript. In response to your questions, we are committed to enhancing the clarity and precision of our work by implementing the following modifications and additions to section 2.1.
In our first version of the manuscript, we wrote the original first paragraph of the Materials and Methods section, specifically in lines 169 to 173 from the original draft (not the revised one):
"First of all, regarding sample collection and processing, cotyledon samples obtained from distinct germination stages, specifically including the imbibition (IMB) stage (designated as number 1), and subsequent days post-imbibition (DAI) stages - 2, 3, 4, 5, 7, 9, and 11 (designated as numbers 2 to 8) - were collected and prepared for microscopy studies."
You are completely right when you point out that the Imbibition process had not been described correctly, and that the "Days After Imbibition" (DAI) were not clearly indicated in the original manuscript. That is the reason why, in this revised manuscript version, and understanding the importance of clarifying this point, we have changed and addressed all of that.
Thus, on our new revised manuscript, we have explicitly stated that our seeds undergo a 24-hour of imbibition period. As such, we have chosen to designate (a maintain) the initial stage of seed development, occurring during the first 24 hours, as "Imbibition" (stage 1). Subsequently, stages 2 to 8 will be consistently referred to as "Germination" to avoid any potential confusion, and will correspond to the different days after imbibition (DAI), or days of germination, that are concretely pointed in the new paragraph.
To address your valuable concern and provide additional clarity, we have then included this new paragraph in lines 204-220, page 4, in yellow color and control/track changes, that elaborates on the imbibition process followed by germination. This section will specifically highlight the initial seed hydration during the "Imbibition" stage, effectively addressing your question and ensuring a more precise description of the seed development process.
“2.1.1. Imbibition and germination of NLL seeds (Lupinus angustifolius L.).
The species used for the conducted studies was L. angustifolius, and the seeds were provided by The University of Western Australia. Seed imbibition was achieved by hydrating the seeds for 24 hours on sufficiently moist filter paper. After this, the seeds were allowed to germinate in MS medium (Murashige and Skoog Medium, specifically used for the germination of NLL to compare germination with and without this medium) or in H2O under sterile conditions. MS medium contains the following mineral salts: NH4NO3 0.47g/l, KNO3 1.31g/l, MgSO4.7H2O 0.24g/l, KH2PO4 0.13g/l, KCl 0.074g/l, Ca(NO3)2.5H2O 0.59g/l, NaFe-EDTA 10ml/l, Micronutrients: MnSO4.H2O 1.690g/l, ZnSO4.7H2O 0.860g/l, CuSO4.5H2O 0.0025g/l, KI 0.0830g/l, CoCl2.7H2O 0.0025g/l, H3BO3 0.620g/l, Na2MoO4.2H2O 0.025g/l, Sucrose 2g/l, and myo-inositol 0.1g/l.
Photographs of the seeds were taken every 24 hours, and a portion of this material was stored at -80°C for subsequent analysis. The final stage studied was 240 hours after imbibition. The stages studied during germination included 24 hours of Imbibition (stage 1,1 DAI), 24 hours of germination (stage 2, 2 DAI), 48 hours (stage 3, 3 DAI), 72 hours (stage 4, 4 DAI), 96 hours (stage 5, 5 DAI), 144 hours (stage 6, 7 DAI), 192 hours (stage 7, 9 DAI), and 240 hours (stage 8, 11 DAI).”
Your input has been invaluable in improving the accuracy and comprehension of our manuscript, and we are grateful for your thoughtful suggestions. These modifications will be diligently incorporated into the revised version of our paper. Thank you once again for your constructive engagement with our work.
6) "NLL seed germination process" should be changed to "NLL seed germination and seedling development process".
Response:
Thank you very much for your valuable feedback. We appreciate your time revising this manuscript, and, in response to your question, we will modify the text to refer to the process as the "NLL seed germination and seedling development process" to accurately represent the scope of our study.
You will find these changes across the whole revised manuscript, always with control or track changes.
7) Please describe the germination conditions of the seeds.
Response:
Thank you very much for your valuable feedback. We appreciate your thorough review of our manuscript. In response to your question, the germination conditions of the seeds will be described in more detail in the revised version of the manuscript, including the duration of imbibition, environmental conditions, and growth medium.
Concretely, and following your comment, we have added a paragraph lines 204-220, page 4, in yellow color and control/track changes, of the revised version of the manuscript, entitled and containing the following:
“2.1.1. Imbibition and germination of NLL seeds (Lupinus angustifolius L.).
The species used for the conducted studies was L. angustifolius, and the seeds were provided by The University of Western Australia. Seed imbibition was achieved by hydrating the seeds for 24 hours on sufficiently moist filter paper. After this, the seeds were allowed to germinate in MS medium (Murashige and Skoog Medium, specifically used for the germination of NLL to compare germination with and without this medium) or in H2O under sterile conditions. MS medium contains the following mineral salts: NH4NO3 0.47g/l, KNO3 1.31g/l, MgSO4.7H2O 0.24g/l, KH2PO4 0.13g/l, KCl 0.074g/l, Ca(NO3)2.5H2O 0.59g/l, NaFe-EDTA 10ml/l, Micronutrients: MnSO4.H2O 1.690g/l, ZnSO4.7H2O 0.860g/l, CuSO4.5H2O 0.0025g/l, KI 0.0830g/l, CoCl2.7H2O 0.0025g/l, H3BO3 0.620g/l, Na2MoO4.2H2O 0.025g/l, Sucrose 2g/l, and myo-inositol 0.1g/l.
Photographs of the seeds were taken every 24 hours, and a portion of this material was stored at -80°C for subsequent analysis. The final stage studied was 240 hours after imbibition. The stages studied during germination included 24 hours of Imbibition (stage 1,1 DAI), 24 hours of germination (stage 2, 2 DAI), 48 hours (stage 3, 3 DAI), 72 hours (stage 4, 4 DAI), 96 hours (stage 5, 5 DAI), 144 hours (stage 6, 7 DAI), 192 hours (stage 7, 9 DAI), and 240 hours (stage 8, 11 DAI).”
8) Microscopy studies need to be clarified. What brand of microscope did you use? How did you stain the tissue? Or give a reference to the staining procedure.
Response:
For microscopy studies, we used a Nikon C1 confocal laser scanning microscope (Nikon) equipped with an argon laser (488nm). Additionally, we stained the tissue with a mixture of 0.05% (w/v) methylene blue and 0.05% (w/v) toluidine blue for 5 minutes at room temperature, followed by a 2-minute wash with running water. These details will be included in the Materials and Methods section, and a reference to the staining procedure will be provided.
We believe that these revisions will enhance the clarity and completeness of our manuscript, and we thank you again for your constructive feedback. We have added two complete paragraphs regarding all of your concerns for the microscopy studies in lines 237-257, page 5, in yellow color and control/track changes,
“2.1.2 Preparation of NLL seed samples for light microscopy
Following the germination of blue lupin seeds, small cotyledon fragments were dissected and fixed for 24 hours at 4°C using a fixation solution. Subsequently, the samples underwent a dehydration process using alcohol solutions with progressively increasing concentrations of 50%, 70%, 96%, 100%, and finally xylene. After dehydration, the samples were embedded in Unicryl plastic resin (BBInternational) at increasing concentrations of Unicryl/ethanol and then subjected to polymerization using exposure to ultraviolet light at -20°C for 72 hours. Semithin sections (≈1μm) of these samples were obtained using a Reichert-Jung Ultracut E ultramicrotome (Leica Microsystems). The semithin sections were placed on glass slides coated with BioBondTM adhesive film (BBInternational).
2.1.3 Observation of NLL seed samples with light microscopy
For general sample observations, the sections were stained with a mixture of 0.05% (w/v) methylene blue and 0.05% (w/v) toluidine blue. In both cases, the sections were stained for 5 minutes at room temperature and washed with running water for 2 minutes. The samples were observed using a Nikon C1 confocal laser scanning microscope (Nikon) with an argon laser (488nm) in bright-field mode. The images were captured and processed using EZ-C1 Gold software version 2.12 build 240 (Nikon).
In order to discern the distribution of lipids within the cotyledon samples, Nile red staining was employed. This lipid-specific staining allowed for the visualization of li-pid-rich regions, shedding light on potential lipid accumulation and mobilization during different germination stages.”
Section 2.2
9) You described the “Oxidative gene expression analyses during the NLL seed germination process”. But no description of the expression analysis of the conglutin family genes.
Response:
Thank you for your keen observation and for taking the time to review our manuscript thoroughly. We appreciate your feedback, and we agree that providing a comprehensive description of the expression analysis of the conglutin family genes is crucial. In light of your suggestion, we have changed the title of this part of Material and Methods (line 274, page 5, yellow color, track changes) to:
“2.2. Gene expression analyses during the NLL seed germination process”
In this part, we describe the general method of study of gene expression in our paper, and then, we have also included 2 different tables (Table 1 and Table 2), where the different primers are listed and described, for both the analysis of gene expression related to oxidative metabolism and the analysis of gene expression of conglutins during the NLL seed germination process.
Both of those tables can be found on section 2.22, lines 380-391, pages 6-7 of the new version of the manuscript, in yellow color and control changes:
Primer |
Forward |
Reverse |
CAT |
GGA ACT ATC CCG AGT GGA AAC |
CCT CAG GCC AAG TCT TAGT TAC |
GsT |
GGA CCC AAA TGA TGG AAC AGA |
GCC AAA CCC AAG TCA ACA AC |
GS |
CAC TAC CAC CAC CAC TCA AA |
TGC GAG GTT CAC GGA TTT |
GPx |
CAA GGA TGA TGC GGA GTA TGT |
TGA AAC CTC CTG TGC CAT AAA |
GR |
GGA GCC AAG GTT GGG ATT T |
GGG AAC ACA GCC ACG AAT AA |
Cu/Zn-SOD |
GGG TCA CCT GGG AAA CAT AG |
CCA CTA AGG CTC TTC CAA CTA C |
NOS |
CCA GAG GTT TGC CTC AGA TT |
TTC ACC AGA TGA ACG GAT TG |
MDHAR-chl |
GAA ACC TAT CCG GTG TTC ACT AT |
TTC ACC TCC AAC AAC TAC AAC |
MDHAR_cyt |
GGA CGA GAG CAG ATT TCC ATA A |
AAA GGA GAA GGG AAA GTG TGA G |
DHAR-chl |
CTC CTC CTT CCC AAC CAT TT |
TTC CTC CAG TGT CAG CAA TAC |
DHAR_cyt |
GAG ACA AGG CTG AGG GTA TTT |
GGA GAT AAG TCC AGA AGG GAA AG |
GAPDH |
CGT GTC CCT ACA GTT GAT GTT |
CCT CCT TGA TGG CAG CTT TA |
“Table 1. Primers for the Analysis of Gene Expression Related to Oxidative Metabolism”
“Table 2. Primers for the Analysis of Gene Expression of Conglutins.”
Primer |
Forward |
Reverse |
α1 |
GAA CAA GGG CAA GGT CCT A |
CAC CAG TAG GCA CTG CTA |
α2 |
GAA GAA ACA AGT ACT CGA G |
GTT CTT GRC CTT GTT CTT G |
α3 |
CAC GGC GAA GAC ATA CAC G |
CTC FFT FFF FAT TFT TCC T |
β1 |
AGG AAT ACG AGC AAG GAG AGG |
GAA AGT GGA TGA CCT GCC G |
β2 |
AAG AAG GGG TTT ATC TTT TCC |
CAT GAT AGT AGA TAG CAG CTT AG |
β3 |
GGA ACA GAG GGA AGA GAG AGA AC |
CAT TGG CTC GTC ATC TTC CTG C |
β4 |
CAG AAG TTA AAG GGC TCA |
TTG TTG TTG TTG GGG T |
β5 |
CAG AGT GGT GAT GAG AGA AAG AC |
CAT TGT CTG CCT GAG CT |
β6 |
GAG CGT GAT CGT GAG CCT T |
GTT CTC TCT CTT CCC TCT GTT CC |
β7 |
GTG AGA TTT CCA ACC TTA GTC |
TTG TTT ACG CGG ACG TG |
γ1 |
ATG GCA CGA AAT ATG GCT CA |
AGA AGT GGG TAG AGG ATA ATG C |
We believe that this addition will enhance the clarity and completeness of our manuscript, allowing readers to better comprehend our research approach. Once again, we thank you for your constructive feedback, which has contributed to the refinement of our work.
10) No information on statistical analysis (number of replicates, significant differences between means).
Response:
Thank you for your review and for highlighting the importance of providing detailed information on statistical analysis in our manuscript. We fully recognize the significance of transparency in reporting statistical methods. In response to your valuable feedback, we have incorporated a dedicated section in the 'Material and Methods' titled '2.4. Statistical Analysis.'
In this section, we now elucidate the number of replicates employed in our experiments and detail the methods used to determine significant differences between means. These additions aim to ensure that readers have a clear understanding of the statistical rigor of this study.
This new section includes the following paragraph, that you will be able to locate at page 8, line 447.454, in yellow color and with control changes:
“2.4 Statistical Analysis
All experiments were performed at least in duplicates or triplicates and the results were expressed as mean ± standard deviation unless otherwise indicated. Statistical analyzes were performed using the Shapiro-Wilk test to analyze the normality of the data set and the One or Two-Way Anova analysis, with Dunnett or Tukey correction, depending on the number of groups and data of each experiment, using Graphad Prism 9 software, version 9.3.0. Statistical differences between samples were considered significant when p values were p < 0.05 (*), p < 0.01 (**), or p < 0.001 (***).”
Once again, we express our sincere gratitude for your thoughtful feedback and the time you have devoted to reviewing our work. Your input has significantly contributed to the enhancement of our manuscript.
11) No information on bioinformatic analysis (software, databases, primer design, DNA sequence).
Response:
Thank you very much for your valuable input. Your observation is of high relevance for our manuscript. Based on your recommendation, we have made significant improvements to the 'Material and Methods' section by including a new paragraph titled 'Bioinformatic Analysis.' In this section, we now provide comprehensive information on the bioinformatic tools, software, databases, primer design, and DNA sequence analysis methods employed in our study.
We believe that these additions enhance the completeness and rigor of our paper, ensuring that readers have a clear understanding of our bioinformatic approach
The new paragraph regarding the bioinformatic analysis appears now in lines 455-592, page 8/9:
“2.5 Bioinformatic Analysis
2.5.1. Genetic resources.
The genes analyzed for expression were sourced from the recently sequenced tran-scriptome of Lupinus angustifolius L. This sequence data was acquired through a col-laborative research effort between two Australian institutions: The University of Western Australia and The Commonwealth Scientific and Industrial Research Organisation (CSIRO). Identification of all the analyzed genes was accomplished using the BLAST tool, comparing them with genes from the same families in model organisms such as Ara-bidopsis and Medicago.
2.5.2. Primer design.
Primers for qPCR assays were designed using the bioinformatic tool PrimerQuest. The design process took into consideration factors such as base pair length similarity, per-centage of GC pairs, and hybridization temperature. These primers were subsequently synthesized by Invitrogen (Thermo Fisher).Once again, we sincerely thank you for your constructive feedback, which has undoubtedly contributed to the refinement of our manuscript.”
Section 2.3
12) Line 163: “anti-β-conglutin antibody from Agrisera was utilized as the primary antibody”. Line 207: “The immunodetection of β-conglutin proteins was carried out through incubation with a rabbit polyclonal antiserum developed in-house”. Please describe the immunocytochemistry protocol in more detail.
Response:
Thank you for your valuable feedback. Indeed, section 2.3 seemed to generate some confusion regarding the biochemical analysis of conglutins during NLL seed germination. To address these concerns and for shake of clarity on the antibody-related aspects and various steps of the immunocytochemistry protocol, we have decided to replace section 2.3. However, we have retained the same section title.
The updated content can now be found in the revised manuscript, specifically in the paragraph spanning lines 409-446, on pages 7 to 8. We have highlighted this new paragraph in yellow and tracked the changes made. We hope that this revision effectively resolves any uncertainties you may have had.
The new paragraph is:
“2.3. Biochemical analysis of β-conglutin proteins during NLL seed germination process.
The protein profile of β-conglutin proteins during the seed germination process of Nar-row-Leafed Lupin (Lupinus angustifolius) was investigated through a series of well-defined biochemical steps.
-SDS-PAGE separation: Initially, protein samples containing 10 µg of protein per sample were subjected to electrophoresis on 4–20% Mini-PROTEAN® TGX™ Precast Gels (Bio-Rad), utilizing the Mini-PROTEAN® Tetra Cell apparatus (Bio-Rad). Fol-lowing electrophoresis, the proteins were visualized using Coomassie Brilliant Blue staining following established protocols.
-Transfer onto PVDF membrane: To facilitate further analysis, the proteins were subsequently transferred from the gel onto polyvinylidene fluoride (PVDF) mem-branes using the Mini-Trans-Blot Electrophoretic Transfer Cell system (Bio-Rad). This step allowed for the subsequent immunodetection of specific proteins.
- Blocking and primary antibody incubation: The PVDF membranes were blocked for 2 hours with a blocking solution containing 5% (w/v) non-fat dry milk in Tris-buffered saline (TBS) buffer at pH 7.4. The immunodetection of β-conglutin proteins was achieved by incubating the membranes with a rabbit polyclonal antiserum developed in-house. The antiserum was diluted to a ratio of 1:1000 in TBS buffer containing 5% (w/v) non-fat dry milk and 0.5% Tween-20.
- Secondary antibody and signal detection: Following primary antibody incubation, a secondary antibody, horseradish peroxidase (HRP)-conjugated anti-rabbit IgG (Bio-Rad), was utilized at a dilution ratio of 1:3000 in TBS buffer with 0.5% Tween-20. The secondary antibody was incubated for 2 hours, followed by three 15-minute washing steps with TBS containing 0.5% Tween-20 to remove any unbound antibodies.
- Chemiluminescence detection: The presence of β-conglutin proteins was detected through chemiluminescence. The membranes were exposed to the SuperSignal® West Pico Chemiluminescent substrate (Thermo Scientific), and the resulting chemiluminescent signal, indicative of the presence of β-conglutin proteins, was captured on X-ray films (Kodak). This comprehensive procedure allowed for the detailed examination of β-conglutin proteins and their variations throughout the Narrow-Leafed Lupin seed germination process.
This biochemical analysis provided crucial insights into the dynamics of β-conglutin proteins during seed germination, further enhancing our understanding of this intricate biological process.”
Result and Discussion:
13) Figure 1: Please improve the resolution and add some points to the caption (DAI, PB, EPB, H). Fig1 (C) – no arrows to indicate lipid droplets. You have shown 11 stages (left) but there are only 6 photos for each staining panel. Indicate at which stage you did the staining.
Response:
Thank you for your valuable feedback regarding Figure 1. We have taken your comments into consideration and made the following improvements. First, we have enhanced the resolution of Figure 1 to provide clearer images for better visualization and interpretation. The new figure appears on line 643, page 10 of the revised version of the manuscript.
We have also revised the caption to include more informative details. The updated caption for Figure 1 now appears at lines 644-658, in yellow color, with track changes, and reads as follows:
“Figure 1. Microscopy study of NLL cotyledon during seed germination. A) and B) Microscopy images obtained from semithin 1μm sections of NLL cotyledon samples from different germination stages (such as IMB, 2, 5, 7, 9, and 11 DAI, respectively), after a general staining using 0.05% (w/v) methylene blue and 0.05% (w/v) toluidine blue solution. At IMB stage, PBs are full of storage proteins. Thenceforth, PBs empty their contents leaving holes (H) by degradation of proteins used as nutrients for plant germination and growth, forming protein (PR) remains located at the PBs periphery. Holes increase their volume creating empty protein bodies (EPB) that fusion themselves until complete disappearance. At this stage, cellular remains (CR) can be observed inside the cells. About 9-11 DAI, only cell wall (CW) is present. C) Microscopy images of 1μm lupin cotyledon sections stained with lipid-specific Nile Red staining. Arrowhead=lipid drop. E) Immunocytochemical localization of β-conglutins and D) control assay. It is demonstrated that β-conglutins co-localize with PBs and the remains they form after its degradation. CR: cellular remains; CW: cell wall; EPB: empty protein bodies; H: hole; PB: protein body; PR: periphery protein bodies remains; Scale bar: A panel: 100µm; B-E panels: 50µm.”
To address your concern about the number of images and the stages at which staining was performed, we have included the pertinent clarification in the caption, before discussing the results, concretely, in lines 644-646:
“1μm sections of NLL cotyledon samples from different germination stages (such as IMB, 2, 5, 7, 9, and 11 DAI, respectively)”
We hope these enhancements address your concerns and provide a clearer representation of the experimental procedures and results in Figure 1.
14) Figure 2: Indicate which antibody was used.
Response:
As previously mention in this response, and already included on the Material and Methods section, the antibody used was:
In order to be clear throughout the document and enable a good lecture from readers, we have added this information in the description and title of Figure 2.
The new title for Figure 2 appears now, in the revised manuscript, on line 763-768, page 12, in yellow color and control changes activated:
“Figure 2. Biochemical analysis of β-conglutins during NLL seed germination process. Upperpanel: proteins profile by SDS-PAGE; Lower panel: immunoblotting using aspecific antibody against β-conglutins. 1: day of imbibition (IMB), 2-8: subsequent days post-imbibition (DAI) stages - 2, 3, 4, 5, 7, 9, and 11 respectively, designated as numbers 2 to 8).The primary antibody used was anti-β-conglutin antibody developed in house (cutom made, Agrisera) following Jimenez-Lopez et al [39], a rabbit polyclonal anti-serum.”
15) Figure 3: Dot bars look like black.
Response:
Thank you very much for your comment. Regarding Figure 3, Black bars and Grey bars do correspond with their respective aspect on the figure but, indeed, dot bars look like white or very light grey. We have consequently changed the description of Figure 3 in order to be coherent with the image.
The new title for Figure 3 appears now, in the revised manuscript, on line 803-810, in yellow color and control changes activated:
“Figure 3. Expression analysis of conglutin families α, β, γ and δ genes during seed germination. The housekeeping gene used as a control was ubiquitin (UBQ). Numbers in X-axis represent days after imbibition (DAI, numbers 2-8). Black bars: imbibition (number 1) and grey bars (seed germinated without MS media). White/light greys bars: seed germinated in MS medium. In all of the graphs, all grey and white/light grey bars (corresponding to the expression of the different genes in different stages post-imbibition with and without MS media) exhibit statistically significant differences (*: p<0.05) com-pared to their respective imbibition stages (black bars, stage 1).”
16) Figure 4: housekeeping gene?
Response:
Thank you for your valuable feedback. As you have correctly pointed out, the original manuscript did not have the specification for the housekeeping gene in Figure 4. In order to address your comment, we have performed two changes. The first one is to add in the Figure 4 title and specification, the details about its housekeeping gene, in this case, the gene used as a control was Glyceraldehyde-3-phosphate de-hydrogenase or GAPDH.
The new title of Figure 4 appears in the revised version of the manuscript, in lines 881-891, in yellow color and track changes, as:
“Figure 4. Expression of oxidative stress-related genes during NLL seed germina-tion. The housekeeping gene used as a control was Glyceraldehyde-3-phosphate de-hydrogenase (GADPH). Genes analyzed correspond to the AsA-GSH cycle as the head of the antioxidant enzyme system such as GPx: Glutathione peroxidase, γ-GCs: γ-Glutamyl-cysteine synthase, GS: Glutathione synthetase, GR: Glutathione reductase, GsT: Glutathione S-transferase; MDHAR: Monodehydro-ascorbate reductase, DHAR: Dehydro-ascorbate reductase; as well as NOS: Nitric oxide synthase, CAT: catalase, Cu/Zn-SOD: Cu/Zn Superoxide dismutase. cyt: cytoplasm, chl: chloroplast and mit: mitochondria. In all cases (measured enzymes) and at every stage, the values exhibit significant differences (*: p<0.05) compared to the black bar, representing the imbibition state, except for Cu/Zn-SOD at stage 2, GS at stages 2 and 7, and DHAR cyt at stage 5 where no statistically significant differences were found”.
Then, and regarding the Materials and Methods sections, we have also added a new paragraph or subsection in line 401-408, page 7, in yellow color and control changes, that also gives answer to your concern:
“2.2.4 Data Analysis
Changes in gene expression levels (shown in Tables 1 and 2, respectively), were determined using the formula 2^-Δ(ΔCt) [46], where the cycle threshold (CT) at which transcripts were detected was normalized to the CT at which the constitutive gene Glyceraldehyde-3-phosphate dehydrogenase (GAPDH) or Ubiquitin (UBQ) was detected, referred to as ΔCT. The PCR efficiency was determined through analysis based on standard curves for the amplification of the selected genes and the endogenous control, which exhibited high reproducibility”.
The reference added in this case still corresponds to:
[46]: Livak KJ, Schmittgen TD. Analysis of relative gene expression data using real-time quantitative PCR and the 2(-Delta Delta C(T)) Method. Methods. 2001 Dec;25(4):402-8. doi: 10.1006/meth.2001.1262.
We appreciate your valuable feedback, and this addition clarifies the data analysis procedures undertaken in our study, Again, thank you very much for all the suggestions you made.
17) Figure 3 and 4: Improve the quality of the figures. Indicate the number of replicates and the confidence level (t-test or ANOVA).
Response:
Thank you very much for your comment. The quality of both images has been improved and, regarding the statistical details, as we previously discussed in section 2.4, this particular paragraph has been added to the Material and Methods section, in order to clarify this point.
The paragraph appears on lines at page 8, line 447-454, in yellow color and with control changes:
“2.4 Statistical Analysis
All experiments were performed at least in duplicates or triplicates and the results were expressed as mean ± standard deviation unless otherwise indicated. Statistical analyzes were performed using the Shapiro-Wilk test to analyze the normality of the data set and the One or Two-Way Anova analysis, with Dunnett or Tukey correction, depending on the number of groups and data of each experiment, using Graphad Prism 9 software, version 9.3.0. Statistical differences between samples were considered significant when p values were p < 0.05 (*), p < 0.01 (**), or p < 0.001 (***).”
Once again, we express our sincere gratitude for your thoughtful feedback and the time you have devoted to reviewing our work.
18) Please divide the Results and Discussion sections. The discussion needs to be extended. It would be good to compare the data on conglutin gene expression with the data obtained by Foley ea 2015.
Response:
We appreciate your thoughtful comments and suggestions regarding the organization of our manuscript. After careful consideration, we would like to provide a rationale for keeping the “Results and Discussion” sections combined in our paper.
First of all, our study involves a comprehensive analysis with extensive datasets, including gene expression data and various experimental results. Combining the “Results and Discussion” sections allows us to present and interpret the data more cohesively and coherently. Separating them could lead to redundancy, as we would need to reiterate the findings in the “Results” section when discussing them in the subsequent discussion section. Then, we also believe that combining the Results and Discussion sections provides a more reader-friendly experience. It allows readers to immediately see the interpretation and significance of the results as they are presented, facilitating a better understanding of the context and implications. Finally, we also wanted to maintain some consistency with our previous publications, as in our recent published articles in journals with similar editorial policies, such as (https://pubmed.ncbi.nlm.nih.gov/37108842/), we followed the same approach of combining Results and Discussion sections, and this approach received positive feedback from both readers and editors.
Regarding your suggestion to extend the “Discussion” section, we have included a more in-depth analysis of conglutin gene expression data, specifically comparing it with the findings of Foley et al. 2015, as you suggested. The expanded discussion points can be found in the revised manuscript from lines 841-876, in control changes and yellow color:
“Concretely, it is very interesting to compare the obtained results of our comprehensive analysis of conglutin gene expression during NLL seed germination with the existing knowledge in this field. Foley et al. (2015) previously investigated conglutin gene expression, and by comparing their findings to our results, several intriguing distinctions emerged [38]. One prominent difference between both results is evident in the expression dynamics of α-conglutin isoforms. While Foley et al. reported a general decrease in α-conglutin expression during germination, our results indicated nuanced variations among α-conglutin isoforms. Specifically, isoform α1-conglutin exhibited higher expression during the imbibition stages compared to α2 and α3 isoforms, which experienced a drastic reduction in expression from 5 DAI. This contrasting pattern may be attributed to phylogenetic and functional differences among α-conglutin isoforms [51]. Furthermore, similar expression trends were observed in the γ-conglutin family, with γ1 and γ2 isoforms showing decreased expression as germination progressed. However, γ2 conglutin displayed higher expression during the early germination stages but decreased by 6 DAI compared to the γ1-conglutin isoform. Additionally, the 4δ-conglutin isoform exhibited a reduction in expression by 5 DAI, while δ1-conglutin decreased at 7 DAI. These disparities in α and γ-conglutin expression dynamics highlight the intricate regulatory mechanisms governing conglutin gene expression during NLL seed germination[38].
Another noteworthy distinction was observed in the β-conglutin family, where our findings revealed diverse transcriptional patterns among isoforms. In contrast to the conventional trend of storage protein degradation during germination, most β-conglutin isoforms in our study exhibited decreased expression levels. However, the standout exceptions were β4 and β6 conglutins, which displayed increased expression levels at later germination stages. Notably, β4 surpassed the 5-day mark after imbibition (DAI), while β6 peaked around 3 DAI. These atypical expression patterns deviated from Foley et al.'s observations and underscore the unique functional and physiological roles that specific β-conglutin isoforms may play during the germination process[38].
While our study aligns with some aspects of Foley et al.'s data, such as the general trend of storage protein degradation, it also elucidates previously unrecognized complexities in conglutin gene regulation during NLL seed germination. These differences underscore the need for further investigation to fully unravel the functional significance of conglutins in the context of seed germination and seedling development.”
Once again, we sincerely thank you for your valuable feedback, which has contributed to improving the quality and clarity of our manuscript.
Conclusion:
19) Line 504: “The significance of the β- and γ-conglutin families as functional signaling molecules in germination progression has been illuminated by this study”. This is speculation. You have shown the dynamics of conglutin mobilization, expression of conglutin genes and ROS-dependent genes. These are very interesting data. But the functional link between them and the signaling role needs to be proven. Please also consider the role of hormonal regulation in these processes.
Response:
We really appreciate your comments on this point and agree that it's essential to substantiate our claims regarding the functional roles of β- and γ-conglutin families in germination progression. While our study has provided valuable data on conglutin mobilization, gene expression patterns, and ROS-dependent genes, we acknowledge that further research is needed to establish the precise functional links between these elements and the signaling roles of conglutins, and in order to fully illuminate the role of this conglutins as functional signaling molecules in germination progress, although our data could in fact suggest that. Additionally, we recognize the importance of considering hormonal regulation in these processes, which can provide a more comprehensive understanding of seed germination, but, as we mentioned in the introduction questions, the inclusion of hormonal regulation would certainly expand the scope of our paper, and while it is a noteworthy area of investigation, it may not be essential for the specific research questions we are addressing in this paper. However, we highly value your input, and we will certainly consider your suggestion for future research or discussions related to this topic, so we will address these aspects in our revised conclusion and future perspective.
Based on all of your comments, we have re-writed the conclusions and future perspective chapter, that now appears in lines 994-1023, in yellow color and with track changes:
“This study has presented compelling insights into the dynamics of conglutin mobilization, their gene expression patterns, and ROS balance-dependent genes during NLL seed germination. These findings shed light on the intricate molecular events that underlie this critical phase of plant development. However, we acknowledge the need for further research to establish the functional connections between these components and the specific signaling roles of β- and γ-conglutin families. To bridge this gap, future investigations will focus on experimental validation to confirm the functional significance of conglutins as signaling molecules during germination progression. Moreover, we will explore the interplay between conglutins and hormonal regulation to gain a more comprehensive understanding of their interactive roles in seed germination.
Beyond the scope of germination, our research opens avenues for broader explorations into the impact of conglutin protein families on subsequent stages of plant development. Investigating their contributions to morphogenesis and developmental processes will provide a holistic view of their influence on the plant life cycle. Furthermore, delving into conglutin interactions within the plant and its environment will uncover their roles in stress responses and adaptive strategies, contributing to our understanding of plant re-silience.
In conclusion, this study serves as a foundation for future researches aimed at substan-tiating the functional significance of NLL conglutin proteins, particularly these from β- and γ- families in germination and unraveling their broader implications for plant growth, development, and adaptability.”

Reviewer 3 Report
In this article, the authors investigated the association between storage proteins mobilization and redox signaling in Narrow-Leafed Lupin seed germination and seedling development. Totally, it is an interesting and valuable study. However, there are a few questions need to be corrected before publication. All detailed comments are as follow:
1. For RT-qPCR, authors should state how many biological duplications were used.
2. For Figure 3 and Figure 4, the difference significance analysis should be performed.
3. Abbreviation should be used throughout the manuscript.
4. Please check the structure of format presentation for “Genes”.
Minor editing of English language required
Author Response
Response to Reviewer 3
Thank you very much for your detailed and precise report about our paper, in where you have been able to identify some issues that have considerably improved the quality of our report. We have taken every detail into account and we will now proceed to respond individually to every mistake that you have identified, in order to facilitate the second revision of our paper. Thank you very much again for your time and revisions and we hope we have been able to respond and modify to every one of your requests.
In this article, the authors investigated the association between storage proteins mobilization and redox signaling in Narrow-Leafed Lupin seed germination and seedling development. Totally, it is an interesting and valuable study. However, there are a few questions need to be corrected before publication. All detailed comments are as follow:
- For RT-qPCR, authors should state how many biological duplications were used.
Response:
Thank you for your valuable input. Your observation regarding the clarification of the number of biological duplications for RT-qPCR is highly appreciated. We have taken your suggestion into account and have now included this essential information in our manuscript to enhance transparency. You can locate the pertinent details in a new paragraph. In this section, we specify that we conducted triplicate reactions for the genes under study for each condition, and all experiments were performed in duplicate or triplicate to ensure a sufficient dataset for robust statistical analysis.
The new paragraph on lines 375-378, page 6, in yellow color and with control changes is:
“Additionally, in our experimental design, we ensured sample distribution by triplicating reactions for the genes of interest on each plate and by repeating at least two or three times the experiments in order to ensure a robust dataset for the performance of statistical analysis.”
- For Figure 3 and Figure 4, the difference significance analysis should be performed.
Response:
Thank you for your review. We greatly appreciate your attention to detail in reviewing this manuscript. You are absolutely right; the inclusion of significance analysis in Figure 3 and Figure 4 is crucial for a comprehensive interpretation of the data. Initially, we refrained from incorporating this information directly into the figures to avoid visual clutter. However, following your valuable recommendation, we have now added the significance analysis to both figure captions, in order to not surcharge the graphs, as they are a lot of statistically significant differences in almost every case. So, in order to clearly indicate those differences without surcharging or perturb the comprehension of the figure, we updated the figure captions of both of them to include a description of it.
Page 13, lines 803-811:
“Figure 3. Expression analysis of conglutin families α, β, γ and δ genes during seed germination. The housekeeping gene used as a control was ubiquitin (UBQ). Numbers in X-axis represent days after imbibition (DAI, numbers 2-8). Black bars: imbibition (number 1) and grey bars (seed germinated without MS media). White/light greys bars: seed germinated in MS medium. In all of the graphs, all grey and white/light grey bars (corresponding to the expression of the different genes in different stages post-imbibition with and without MS media) exhibit statistically significant differences (*: p<0.05) com-pared to their respective imbibition stages (black bars, stage 1).”
Page 15, lines 883-893:
“Figure 4. Expression of oxidative stress-related genes during NLL seed germina-tion. The housekeeping gene used as a control was Glyceraldehyde-3-phosphate de-hydrogenase (GADPH). Genes analyzed correspond to the AsA-GSH cycle as the head of the antioxidant enzyme system such as GPx: Glutathione peroxidase, γ-GCs: γ-Glutamyl-cysteine synthase, GS: Glutathione synthetase, GR: Glutathione reductase, GsT: Glutathione S-transferase; MDHAR: Monodehydro-ascorbate reductase, DHAR: Dehydro-ascorbate reductase; as well as NOS: Nitric oxide synthase, CAT: catalase, Cu/Zn-SOD: Cu/Zn Superoxide dismutase. cyt: cytoplasm, chl: chloroplast and mit: mitochondria. In all cases (measured enzymes) and at every stage, the values exhibit significant differences (*: p<0.05) compared to the black bar, representing the imbibition state, except for Cu/Zn-SOD at stage 2, GS at stages 2 and 7, and DHAR cyt at stage 5 where no statistically significant differences were found.”
Additionally, in response to your suggestion, we have included a dedicated paragraph in the Materials and Methods section to elaborate on the statistical analysis procedures employed. This information can be found in the revised manuscript under a new section.
This new section includes the following paragraph, that you will be able to locate at page 8, lines 447-454, in yellow color and with control changes:
“2.4 Statistical Analysis
All experiments were performed at least in duplicates or triplicates and the results were expressed as mean ± standard deviation unless otherwise indicated. Statistical analyzes were performed using the Shapiro-Wilk test to analyze the normality of the data set and the One or Two-Way Anova analysis, with Dunnett or Tukey correction, depending on the number of groups and data of each experiment, using Graphad Prism 9 software, version 9.3.0. Statistical differences between samples were considered significant when p values were p < 0.05 (*), p < 0.01 (**), or p < 0.001 (***).”
Once again, we express our sincere gratitude for your thoughtful feedback and the time you have devoted to reviewing our work. Your input has significantly contributed to the enhancement of our manuscript.
- Abbreviation should be used throughout the manuscript.
Response:
Thank you very much for your valuable comment. We greatly appreciate your feedback, and we concur with the importance of employing abbreviations to enhance the readability and conciseness of the manuscript. Following your suggestion, we have diligently incorporated appropriate abbreviations consistently throughout the entire manuscript.
These changes have been implemented across all sections, and they are clearly highlighted in yellow with track changes enabled for your convenience. Once again, we extend our gratitude for your thoughtful feedback and the time you've dedicated to reviewing our work.
- Please check the structure of format presentation for “Genes”.
Response:
Thank you very much for your comment. We sincerely appreciate the time you've dedicated to reviewing our manuscript. We have carefully reviewed the format and presentation of 'Genes' and believe that with the recent changes, our manuscript aligns with the required format. It now includes an abstract, introduction, materials and methods with their respective sections, results and discussion within the same section to enhance comprehension, followed by a conclusion and a bibliography in accordance with the specified format.
However, if there are any specific structural issues that we may have missed, please do not hesitate to let us know. We are more than willing to make any necessary adjustments to ensure that our manuscript meets the required format standards. Once again, we extend our gratitude for your feedback and your valuable time dedicated to the review of our article.

Round 2
Reviewer 2 Report
The authors carefully reviewed the article and responded to all comments.